# Specific and Shared Causal Relation Modeling and Mechanism-Based Clustering

**Biwei Huang** [1] *, **Kun Zhang**[1], **Pengtao Xie**[2], **Mingming Gong**[3], **Eric Xing**[2,4], **Clark Glymour**[1]

[1]Department of Philosophy, Carnegie Mellon University, Pittsburgh, PA, USA.
[2]Petuum Inc., USA.
[3]School of Mathematics and Statistics, University of Melbourne, Melbourne, Australia.
[4]Department of Machine Learning, Carnegie Mellon University, Pittsburgh, PA, USA.

## Abstract

State-of-the-art approaches to causal discovery usually assume a fixed underlying causal model. However, it is often the case that causal models vary across domains or subjects, due to possibly omitted factors that affect the quantitative causal effects. As a typical example, causal connectivity in the brain network has been reported to vary across individuals, with significant differences across groups of people, such as autistics and typical controls. In this paper, we develop a unified framework for causal discovery and mechanism-based group identification. In particular, we propose a specific and shared causal model (SSCM), which takes into account the variabilities of causal relations across individuals/groups and leverages their commonalities to achieve statistically reliable estimation. The learned SSCM gives the specific causal knowledge for each individual as well as the general trend over the population. In addition, the estimated model directly provides the group information of each individual. Experimental results on synthetic and real-world data demonstrate the efficacy of the proposed method.

## 1 Introduction

Learning causal relations from observational data automatically, known as causal discovery, has shown its increasing importance and efficacy. State-of-the-art approaches to causal discovery usually assume a fixed causal model [34, 2, 13, 32, 14, 40, 16]; that is, causal mechanisms are invariant across instances in the data set. Under this assumption, causal relations can be identified by leveraging the conditional independence between observed variables [34] or the asymmetrical independence between estimated noise term and hypothetical causes, implied by suitable functional causal models [32, 38, 14, 40].

In real-world scenarios, it is often the case that causal relations over the considered set of variables may vary across individuals or individual groups, and meanwhile they also share many commonalities. For example, in healthcare, individuals may show different responses to the same treatment. The varying responses may be due to some (unmeasured) factors, such as nutrition and health status. At the same time, although the effect may be different for different individuals, a large proportion may still show a similar trend, while others may show very distinct effects. This suggests that to understand causal effects, it is helpful to properly divide these subjects into different groups: within groups, the variation of the treatment effect should be small, while it may be large across groups. When examining whether a treatment is effective and should be adopted as standard practice, one should not only care about its effect in the general population, but also account for the response to the treatment of each individual or each properly divided group. The brain network is another example. There is ample evidence that heterogeneity in brain processes exists across individuals [35].

More specifically, there exist significant differences of brain information flows in different groups of people. For example, it has been shown that cases of autism are associated with atypical brain connectivities [5, 17], and that the differences provide a good criterion for autism diagnosis and help to localize neuropathology biomarkers.

To find the difference across individuals, a typical solution is to analyze data from each subject separately and then make a comparison. However, this approach may suffer from low statistical reliability, because the size of samples from one subject may be small while the data dimension may be high. For example, in healthcare data, each patient may have only a few records due to resource and time constraints, while many clinical variables are measured. Fortunately, although individuals may not have the same causal model, they usually share many commonalities, which can be leveraged to achieve more reliable estimation results. This reasoning is motivated by multi-task learning [1], where multiple learning tasks are solved jointly in a principled way, thus exploiting their commonalities while at the same time preserving useful information for each individual task. On the other hand, if we ignore the differences and concatenate the data to estimate a causal graph, spurious edge or incorrect causal directions may be introduced [39].

Recently, some approaches have been developed for causal discovery in the case where causal relations over the observed variables change across domains. For example, causal discovery from heterogeneous/nonstationary data (CD-NOD) [39, 19] concatenates data from different domains and considers the domain index as a surrogate to characterize the variability of causal mechanisms, and finally recovers fixed as well as varying causal relations. In the linear case, invariant causal relations are found based on invariant predictions [26], and some other methods can directly estimate varying causal relations [18, 20, 11, 37, 15]. Despite their success on the considered problem, these approaches do not explicitly provide the group-level information regarding which ones are similar to each other and can be grouped together. In addition, they do not allow opposite causal directions in different domains. The Group Iterative Multiple Model Estimation approach [23, 9] tries to recover time-lagged causal relations at both group and individual levels in a heuristic way, with no theoretical guarantees. The instance-specific greedy equivalence search [22] outputs a Markov equivalence class that is specific to a given instance $T$ by guiding the search based on $T$'s attributes, but it is only for discrete data.

Motivated by the above real-world scenarios in healthcare and neuroscience, we propose a Specific and Shared Causal Model (SSCM), to achieve the following goals: (1) Discover a general trend of causal relations over the population. (2) Identify specific causal relations for each individual or each automatically determined group. (3) Exploit variations and commonalities of causal relations to cluster individuals into different groups. In particular, for each individual, the causal model is formalized with a linear non-Gaussian model. The learned specific and shared causal model gives the information of the specific causal knowledge for each individual, as well as the general trend over the population. Each individual can be grouped by directly using the learned causal model. Moreover, the proposed causal model is theoretically identifiable under mild conditions.

## 2  Specific and Shared Causal Models

Suppose there are $n$ individuals, which can be divided into $q$ groups; we do not know which group each individual belongs to. All individuals have the same $m$ observed variables under investigation, but their causal relations may be different. For the $s$-th individual ($s = 1, \cdots, n$), we observe $l_s$ data points for the $m$ variables; the $l_s$ data points can be either independent and identically distributed (i.i.d.) or from a stationary time series. Consider the brain connectivity problem. We have $n$ subjects, which are expected to be from $q$ groups; for the $s$-th subject we record $l_s$ fMRI data points over $m$ variables. We aim to learn a shared causal model over the $m$ variables, which is shared across the population, and also a specific causal model for each individual. Moreover, we cluster these $n$ individuals into $q$ groups by leveraging the learned causal model.

Suppose the $m$ observed variables from the $s$-th individual, $X^s(t) = (x_1^s(t), \cdots, x_m^s(t))^{\mathrm{T}}$, satisfy the following generating process

$$x_i^s(t) = \underbrace{\sum_{j \in \mathcal{P}_i^s} b_{ij}^s x_j^s(t)}_{\text{instantaneous}} + \underbrace{\sum_{p=1}^{p_l} \sum_{j \in \mathcal{L}_i^s} a_{ij,p}^s x_j^s(t-p)}_{\text{time-lagged}} + e_i^s(t), \tag{1}$$

for $i = 1, \cdots, m$, where $b_{ij}^s$ represents instantaneous causal influences from variable $j$ to $i$ in the $s$-th subject, $a_{ij,p}^s$ $p$-lagged causal influences, $\mathcal{P}_i^s$ the set of indices of instantaneous direct causes of $x_i^s$, and $\mathcal{L}_i^s$ the set of indices of lagged direct causes of $x_i^s$. Each individual has fixed causal coefficients $b_{ij}^s$ and $a_{ij,p}^s$, while they may change across individuals. The noise term $e_i^s(t)$ is non-Gaussian, representing some unmeasured factors. It is independent of $x_j^s(t)$ and $x_j^s(t-p)$, for all $j, p \in \mathcal{N}^+$. Note that for i.i.d. samples, we only consider instantaneous causal relations, while for stationary time series, we allow both instantaneous and time-lagged causal relations. Eq. (1) can be represented in the matrix form:

$$X^s(t) = B^s X^s(t) + \sum_{p=1}^{p_l} A_p^s X^s(t-p) + E^s(t), \qquad (2)$$

where $B^s$ is the $m \times m$ instantaneous causal adjacency matrix with entries $b_{ij}^s$, $A_p^s$ the $m \times m$ lagged causal adjacency matrix with entries $a_{ij,p}^s$, and $E^s(t) = (e_1^s(t), \cdots, e_m^s(t))^\mathsf{T}$.

We allow that both instantaneous causal relations $b_{ij}^s$ and lagged relations $a_{ij,p}^s$ change across groups or individuals. More specifically, they vary across different groups, while there are also slight differences across individuals within the group. Intuitively, one may estimate corresponding causal relationships for each individual separately. However, the limited sample size from each individual limits statistical efficiency or even makes causal discovery impossible, especially when the data dimension is high and the causal graph is dense. Although individuals may not have the same causal model, they usually share many commonalities, which can be leveraged to achieve more reliable estimation results.

To exploit both variations and commonalities across groups, as well as across individuals, and meanwhile perform mechanism-based clustering in a principled way, we propose *specific and shared causal relation modeling*. Specifically, we take the instantaneous causal influence as a random variable $b_{ij}$, where $b_{ij}^s$ can be seen as an instance of $b_{ij}$. To encode the variation across groups, as well as that within each group, we assume that in each group, $b_{ij}$ follows a Gaussian distribution, while in different groups the Gaussian distributions are different. Therefore, we impose a mixture of Gaussians (MoG) prior on $b_{ij}$. Let a $q$-dimensional binary vector $Z$ denote the index of the Gaussian component in the MoG for $b_{ij}$ (i.e., which group the individual belongs to) where a particular element $z_k = 1$ and all other elements are zero. Then the distribution of $b_{ij}$ can be represented as

$$P(b_{ij}) = \sum_{k=1}^{q} P(z_k = 1) P(b_{ij}|\mu_{k,ij}, \sigma_{k,ij}), \qquad (3)$$

where $P(b_{ij}|\mu_{k,ij}, \sigma_{k,ij}) = \mathcal{N}(b_{ij}|\mu_{k,ij}, \sigma_{k,ij}^2)$ and $\mathcal{N}(\cdot)$ denotes a Gaussian distribution, $P(z_k = 1) = \pi_k$, and $\sum_{k=1}^{q} \pi_k = 1$. Similarly, for lagged causal influences, we have

$$P(a_{ij,p}) = \sum_{k=1}^{q} P(z_k = 1) P(a_{ij,p}|\nu_{k,ij,p}, \omega_{k,ij,p}), \qquad (4)$$

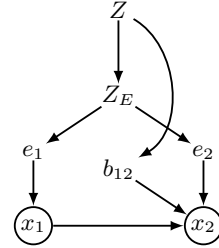

Figure 1: Graphical representation of a two-variable case: $x_1$ and $x_2$ are two observed variables, $b_{12}$ is the instantaneous causal strength from $x_1$ to $x_2$, $e_1$ and $e_2$ denote the noise terms w.r.t $x_1$ and $x_2$, respectively, $Z$ is the group indicator, and $Z^E$ is the indicator of the MoG of $e_1$ and $e_2$.

where $P(a_{ij,p}|\nu_{k,ij,p}, \omega_{k,ij,p}) = \mathcal{N}(a_{ij,p}|\nu_{k,ij,p}, \omega_{k,ij,p}^2)$. Different causal coefficients ($a_{ij,p}$ and $b_{ij}$, for all $i, j, p$) in the same group share the same $P(Z)$.

We also allow the noise distribution varies across groups but remains the same within the group. More specifically, we model the non-Gaussian noise in each group with an MoG, and in different groups, the noise may have different MoG distributions. Denote by $Z^E$ the indicator of the MoG of $E$, with $Z^E = (z_1^E, \cdots, z_{q'}^E)$, and thus in group $k$ with $z_k = 1$, the distribution of $E$ is:

$$P(E|z_k = 1) = \sum_{k'=1}^{q'} P(z_{k'}^E = 1|z_k = 1) P(E|z_{k'}^E = 1, z_k = 1), \qquad (5)$$

where $P(E|z_{k'}^E = 1, z_k = 1) = \mathcal{N}(E|\vec{\mu}_{k,k'}^E, \Sigma_{k,k'}^E)$, $P(z_{k'}^E = 1|z_k = 1) = \pi_{k,k'}^E$, and $\sum_{k'=1}^{q'} \pi_{k,k'} = 1$. Thus, $P(E) = \sum_{k=1}^{q} P(E|z_k = 1) P(z_k = 1)$.

Figure 1 shows the graphical representation of the entire model in a two-variable case, with only instantaneous causal relations. Because in our model we consider $b_{ij}$ as a random variable, there is a causal edge from $b_{12}$ to $x_2$. Therefore, the specific and shared causal model is represented as

$$X(t) = BX(t) + \sum_{p=1}^{p_l} A_p X(t-p) + E(t), \qquad (6)$$

with

$$
\begin{aligned}
P(b_{ij}) &= \textstyle\sum_{k=1}^{q} \pi_k \mathcal{N}(b_{ij}|\mu_{k,ij}, \sigma_{k,ij}^2), \\
P(a_{ij,p}) &= \textstyle\sum_{k=1}^{q} \pi_k \mathcal{N}(a_{ij,p}|\nu_{k,ij,p}, \omega_{k,ij,p}^2), \\
p(E) &= \textstyle\sum_{k=1}^{q} \pi_k \sum_{k'=1}^{q'} \pi_{k,k'}^E \mathcal{N}(E|\vec{\mu}_{k,k'}^E, \Sigma_{k,k'}^E).
\end{aligned}
\qquad (7)
$$

In the next section, we will discuss its identifiability; the identifiability applies to both the causal structure and model parameters.

## 3 Model Identifiability

Theorem 1 shows the identifiability in a specific case, where there are different groups, and causal relations are different across groups but identical within each group.

**Theorem 1.** *The proposed causal model in (6) and (7), including the causal structure and model parameters, is identifiable, as $n \to \infty$, under the following conditions:*

1. *The parameters $\sigma_{k,ij} = 0$ and $\omega_{k,ij,p} = 0$, for all $i, j, k, p \in \mathcal{N}^+$.*

2. *The sample size of each individual $l_s > 2q - 1$, where $q$ is the number of groups.*

3. *The instantaneous causal structure for each individual is acyclic.*

Note that although the instantaneous causal structure for each individual is acyclic, we allow that across different groups, some causal directions are reversed. For instance, in the brain network, different directions may be activated across subjects or states, which is hard to handle with traditional methods. In addition, if there are cycles in the instantaneous causal structure, the identifiability requires two more conditions [24]: (1) the cycles are disjoint, and (2) the causal model is stable, i.e., $\lim_{k \to \infty} B^k = 0$. As an unsupervised method, the order of the group index is not identifiable; i.e., it can be arbitrarily permuted. We are aware that in the above result it is assumed that there is no variation within groups. For the general case, the proof of the identifiability results does not seem immediate, but our empirical results suggest that the causal model is also identifiable. In the following, we give a proof sketch of Theorem 1; for detailed proofs, please refer to the supplementary material.

*Proof Sketch.* Condition 1 means that $b_{ij}$ and $a_{ij,p}$ take a degenerate Gaussian distribution in each group; their distributions can be represented as follows:

$$P(b_{ij}) = \textstyle\sum_{k=1}^{q} \pi_k \delta_{\mu_{k,ij}}(b_{ij}), \quad P(a_{ij,p}) = \textstyle\sum_{k=1}^{q} \pi_k \delta_{\nu_{k,ij,p}}(a_{ij,p}),$$

where $\delta_{\mu_{k,ij}}(b_{ij}) = 1$, if $b_{ij} = \mu_{k,ij}$, and 0 if otherwise; $\delta_{\nu_{k,ij,p}}(a_{ij,p}) = 1$, if $a_{ij} = \nu_{k,ij,p}$, and 0 if otherwise. With condition 1, the identifiability of the proposed causal model can be seen from the view of finite mixture models with grouped samples. The "grouped samples" means that for each individual there are several samples, and it is known in advance that they are identically distributed samples from the same component. Note that the identifiability of finite mixture models with grouped samples is easier to achieve (see [36]), compared to the case where the observations are drawn i.i.d. from a mixture model. Imagine an extreme case: if there are enough samples for each individual, then the corresponding components can be identified from each individual directly.

We first show that under condition 1 and 2, where $l_s > 2q - 1$ [36], the cumulative distribution function of each mixture component, as well as the mixture proportion, is identifiable; that is, $P(X|z_k = 1)$ is identifiable, for $k = 1, \cdots, q$, and $P(Z)$ is identifiable. Next, thanks to the identifiability of independent component analysis-based models [7, 32], we can show that in each group, instantaneous causal relations $b_{ij}$ and lagged causal relations $a_{ij,p}$ are identifiable [21]. $\qquad \square$

# 4 Model Identification

The specific and shared causal model defined above can be regarded as a latent variable model, with $\mathbf{U} = \left\{ \{b_{ij}\}_{i,j=1}^m, \{a_{ij,p}\} \right\}$ as latent variables that we are interested in, and $\theta = \left\{ \{\pi_k\}, \{\mu_{k,ij}\}, \{\sigma_{k,ij}\}, \{\nu_{k,ij,p}\}, \{\omega_{k,ij,p}\}, \{\pi_{k,k'}^E\}, \{\vec{\mu}_{k,k'}^E\}, \{\Sigma_{k,k'}^E\} \right\}$ as free parameters that need to be estimated. In particular, we exploit a stochastic approximation expectation maximization (SAEM) algorithm [4], combined with Gibbs sampling in the E step and EM algorithm in the M step, for model estimation.

## 4.1 Parameter Estimation with SAEM

For a traditional EM algorithm, the procedure is initialized at some $\theta_0 \in \Theta$ and then iterates between two steps, expectation (E) and maximization (M):

(E) Compute $P_{\theta^{r-1}}(\mathbf{U}|X)$ and the lower bound of the log-likelihood, $\mathcal{Q}(\theta, \theta^{r-1})$, with

$$\mathcal{Q}(\theta, \theta^{r-1}) = \int P_{\theta^{r-1}}(\mathbf{U}|X) \log P_\theta(X, \mathbf{U}) \, d\mathbf{U}.$$

(M) Compute $\theta^r = \arg\max_{\theta \in \Theta} \mathcal{Q}(\theta, \theta^{r-1})$.

In the E step, we need to compute the expectation under the posterior $P_{\theta^{r-1}}(\mathbf{U}|X)$, which is intractable in our case, since $P(X, \mathbf{U})$ is not Gaussian. To address this issue, SAEM computes the E step by Monte Carlo integration and uses a stochastic approximation update of the quantity $\mathcal{Q}$ at the $r$-th iteration:

$$\tilde{\mathcal{Q}}_r(\theta) = (1 - \lambda_r)\tilde{\mathcal{Q}}_{r-1}(\theta) + \lambda_r \sum_{j=1}^M \frac{1}{M} \log P_\theta(\mathbf{X}^{1:n}, \mathring{U}^{(1:n,r,j)}), \tag{8}$$

where $\mathring{U}$ indicates sampled particles of $\mathbf{U}$, $M$ is the generated number of particles, $\mathbf{X}^{1:n} = \{\mathbf{X}^s\}_{s=1}^n$ and $\mathbf{X}^s = (X^s(1), \cdots, X^s(l_s))$, $\mathring{U}^{(1:n,r,j)} = \{\mathring{U}^{(s,r,j)}\}_{s=1}^n$, and $\{\lambda_r\}_{r \geq 1}$ is a decreasing sequence of positive step size, with $\sum_r \lambda_r = \infty$ and $\sum_r \lambda_r^2 < \infty$. More specifically, given the learned parameters in the current iteration, the values of latent variables are first sampled under the posteriori density. Then these sampled data are used to update the value of the conditional expectation of the complete log-likelihood with stochastic approximation. The E-step is thus replaced by the following:

(E′) At each iteration, generate $M$ particles of $\mathring{U}^{(1:n,r,j)}$ from $P_{\theta^{r-1}}(\mathbf{U}|X)$ and compute $\tilde{\mathcal{Q}}_r(\theta)$ according to (8). A method for sampling from $P_{\theta^{r-1}}(\mathbf{U}|X)$ is introduced in the following.

**Gibbs Sampling in E-step**  Since the dimension of latent variables $\mathbf{U}$ may be high, especially when $m$ is large, we use Gibbs sampling to sample particles $\mathring{U}$ from the posterior distribution, and within Gibbs sampling, we use independent doubly adaptive rejection metropolis sampling (IA$^2$RMS) [25].

The idea in Gibbs sampling is to generate posterior samples by sweeping through each variable to sample from its conditional distribution with the remaining variables fixed to their current values. At each iteration, perform

$$b_{ij} \sim P(b_{ij}|\mathbf{X}^{1:n}, \mathbf{U}\backslash b_{ij}), \quad a_{ij,p} \sim P(a_{ij,p}|\mathbf{X}^{1:n}, \mathbf{U}\backslash a_{ij,p}), \tag{9}$$

for all $i, j, p \in \mathbf{N}^+$, where $\mathbf{U}\backslash b_{ij}$ and $\mathbf{U}\backslash a_{ij,p}$ denote all variables in $\mathbf{U}$ except $b_{ij}$ and $a_{ij,p}$, respectively. In each sampling, we use IA$^2$RMS. It differs from adaptive rejection metropolis sampling, with an additional adaptive step to improve the proposal probability density function.

**EM Algorithm in M-step**  In the M step, we compute $\theta^r = \arg\max_{\theta \in \Theta} \mathcal{Q}(\theta, \theta^{r-1})$. It is achieved by an inner EM algorithm. See supplementary materials for detailed derivations.

The computational complexity of SAEM in each iteration is $O(m^2 nMT_0)$, where $m$ is the number of variables, $n$ the number of subjects, $M$ the number of sampled particles (we used $M = 30$), and $T_0$ the number of iterations needed in the Gibbs sampling for each variable, depending on the number of rejection times and the number of supporting points that need to be calculated in the adaptive rejection sampling.

## 4.2 Specific and Shared Causal Relation Determination

After estimating the parameters, we can derive the posterior distribution of $\{A_p\}_{p=1}^{p_l}$ and $B$, with

$$P(\{A_p\}, B|\mathbf{X}^{1:n}) \propto P(\mathbf{X}^{1:n}|\{A_p\}, B) \prod_{i,j,p} P(a_{ij,p})P(b_{ij}), \tag{10}$$

where

$$P(\mathbf{X}^{1:n}|\{A_p\}, B) = |\det(I - B)|^{\sum_{s=1}^{n} l_s} \cdot P_E\left((I - B)\check{\mathbf{X}}_0 - \sum_p A_p \check{\mathbf{X}}_p\right),$$

$$P(b_{ij}) = \sum_{k=1}^{q} \pi_k \mathcal{N}(b_{ij}|\mu_{k,ij}, \sigma_{k,ij}^2), \quad P(a_{ij,p}) = \sum_{k=1}^{q} \pi_k \mathcal{N}(a_{ij,p}|\nu_{k,ij,p}, \omega_{k,ij,p}),$$

$$P_E\left((I - B)\check{\mathbf{X}}_0 - \sum_p A_p \check{\mathbf{X}}_p\right) = \sum_{k=1}^{q} \pi_k \sum_{k'=1}^{q'} \pi_{k,k'}^{E} \mathcal{N}\left((I - B)\check{\mathbf{X}}_0 - \sum_p A_p \check{\mathbf{X}}_p | \mu_{k,k'}^{E}, \Sigma_{k,k'}^{E}\right),$$

$$\check{\mathbf{X}}_0 = (\mathbf{X}_{p+1:l_1}^1, \cdots, \mathbf{X}_{p+1:l_n}^n), \quad \text{and} \quad \check{\mathbf{X}}_p = (\mathbf{X}_{1:l_1-p}^1, \cdots, \mathbf{X}_{1:l_n-p}^n).$$

Then the estimated specific causal relationships are implied by the posterior distribution of $A_p$ and $B$ given the data from the $s$-th individual. More specifically, one may take the maximum a posterior (MAP) as a point estimator of $A_p$ and $B$:

$$\{\{\hat{A}_p^s\}, \hat{B}^s\} = \underset{\{A_p\},B}{\arg\max} \, P(\{A_p\}, B|\mathbf{X}^s). \tag{11}$$

The estimated shared causal relationships are implied by the posterior distribution of $A_p$ and $B$ given the data from all individuals, and its point estimator is

$$\{\{\hat{A}_p\}, \hat{B}\} = \underset{\{A_p\},B}{\arg\max} \, P(\{A_p\}, B|\mathbf{X}^{1:n}). \tag{12}$$

Recall that the linear non-Gaussian acyclic model (LiNGAM [32]) first estimates $W = (I - B)^{-1}$ and then recovers the underlying adjacency matrix $B$ by performing extra permutation and rescaling, since $W$ is only identified up to permutation and scale. In our model, we directly model the causal process $B$, with the following advantages: (1) It is easy to add prior knowledge of causal connections. In practice, experts may have domain knowledge about some causal edges. (2) One can directly enforce sparsity constraints on causal adjacencies. (3) The estimation procedure directly outputs the causal adjacency matrix, without additional steps of permutation and rescaling, which are usually expensive.

## 5 Mechanism-based Clustering with Specific and Shared Causal Model

After estimating the specific and shared causal model, we can immediately cluster individuals into $q$ groups, by estimating $P(z_k = 1|\mathbf{X}^s)$, for $k = 1, \cdots, q$, where

$$P(z_k = 1|\mathbf{X}^s) \propto P(\mathbf{X}^s|z_k = 1)P(z_k = 1), \tag{13}$$

and

$$P(\mathbf{X}^s|z_k = 1) = \int \int P(\mathbf{X}^s|\{A_p\}, B, z_k = 1)P(\{A_p\}, B|z_k = 1) \, d\{A_p\} \, dB. \tag{14}$$

The above integration does not have a closed form, and thus we use Monte Carlo integration. We sample $M$ values of $\{A_p\}$ and $B$ from $P(\{A_p\}, B|z_k = 1)$, and thus

$$
\begin{aligned}
&P(\mathbf{X}^s|z_k = 1) \\
= \; &\frac{1}{M} \sum_{i=1}^{M} |\det(I - B^{(i)})|^{l_s} \cdot \sum_{k'=1}^{q'} \pi_{k,k'} \mathcal{N}\left((I - B^{(i)})\mathbf{X}_{p+1:l_s}^s - \sum_p A_p \mathbf{X}_{1:l_s-p}^s; \vec{\mu}_{k,k'}^{E}, \Sigma_{k,k'}^{E}\right),
\end{aligned}
$$

where $A_p^{(i)}$ and $B^{(i)}$ denote the sampled $i$-th value from $P(\{A_p\}, B|z_k = 1)$. Therefore,

$$P(z_k = 1|\mathbf{X}^s) \propto \frac{\pi_k}{M} \sum_{i=1}^{M} |\det(I - B^{(i)})|^{l_s} \cdot \sum_{k'=1}^{q'} \pi_{k,k'} \mathcal{N}\left((I - B^{(i)})\mathbf{X}_{p+1:l_s}^s - \sum_p A_p \mathbf{X}_{1:l_s-p}^s | \mu_{k,k'}^{E}, \Sigma_{k,k'}^{E}\right).$$

Denote by $c^s$ the group that individual $s$ belongs to. The estimated group index for individual $s$ is finally given by:

$$\hat{c}^s = \underset{k}{\arg\max} \, P(z_k = 1|\mathbf{X}^s). \tag{15}$$

# 6 Experimental Results

To show the efficacy of the proposed approach to specific and shared causal relation discovery and its performance in mechanism-based clustering, we apply it to both synthetic and real-world data.

**Synthetic Data**   We randomly generated acyclic causal structures according to the Erdos-Renyi model [6] with parameter 0.3. We denote by $G$ the graph structure. Each generated graph has 5 variables. To show the generality of the proposed method, we varied the number of groups with $q = 2, 3$, the number of samples for each individual with $l_s = 20, 40, 60$, and the number of individuals with $n = 60, 80, 100$. Motivated from the real-world scenario that brain connectivities may be enhanced or inhibited in individuals with mental disorders, such as autism and schizophrenia, compared to typical controls, the parameters were set in the following way:

- In the 2-group case ($q = 2$), when the group index $k = 1$ (e.g., typical control group), we set $\mu_{k,ij} \sim \mathcal{U}(0.8, 1)$ for all $i, j$ where $G_{ij} = 1$; when $k = 2$ (e.g., autism group), we randomly sampled pairs of $i', j'$ where $G_{i'j'} = 1$, and set $\mu_{k,i'j'} \sim \mathcal{U}(0, 0.2)$ to model the situation that some causal edges are inhibited, and for the remaining $i, j$ where $G_{ij} = 1$, $\mu_{k,ij} \sim \mathcal{U}(0.8, 1)$.
- In the 3-group case ($q = 3$), when $k = 1$ or $k = 2$, $\mu_{k,ij}$ was the same as above; when $k = 3$ (e.g., schizophrenia group), we randomly sampled pairs of $i', j'$ where $G_{i'j'} = 1$, and set $\mu_{1,i'j'} \sim \mathcal{U}(1.8, 2)$ to model the situation that some causal edges are enhanced.

Other parameters were set as follows: $\sigma_{k,ij}^2 \sim \mathcal{U}(0.01, 0.1)$, $\omega_{k,ij,p}^2 \sim \mathcal{U}(0.01, 0.1)$, each entry of $\vec{\mu}_{k,k'}^E \sim \mathcal{U}(-0.6, -0.4) \cup \mathcal{U}(0.4, 0.6)$, each entry of $\Sigma_{k,k'}^E \sim \mathcal{U}(0.2, 0.5)$, $\pi_k \sim \mathcal{U}(0.3, 0.6)$, $\pi_{k,k'}^E \sim \mathcal{U}(0.3, 0.6)$, and $\sum_{k=1}^{q} \pi_k = 1$, $\sum_{k'=1}^{q'} \pi_{k,k'}^E = 1$, where $\mathcal{U}(l, u)$ denotes a uniform distribution between $l$ and $u$. For each setting (a particular group size $q$, a particular sample size for each individual $l_s$, and the number of individuals $n$), we generated 30 realizations.

For causal discovery, we identified specific causal relations by the proposed approach. We compared it with other well-known approaches in causal discovery, including LiNGAM [32], the minimal change method (MC) [11], and the identical boundaries method (IB) [11]. In particular, we applied LiNGAM on each individual separately because it assumes a fixed causal model. Both MC and IB leverage the minimal change principle to identify the causal structure. Since the state-of-the-art baselines, such as LiNGAM, MC, and IB, only consider instantaneous causal relations, we report the identification results of instantaneous causal relations. For time-lagged causal relations, the causal direction is fixed (from past to future), and thus, it reduces to a parameter identification problem.

In our method, we initialized the parameters in the following way: we first estimated the correlation matrix for each individual and clustered the estimated correlation matrices with K-means clustering, and then we used the estimated centroids of each group as the initial value of $\mu_{k,ij}$. Other parameters were initialized randomly. In our experiments, the number of groups was given. If there is a large number of groups, one may use some information criteria, such as the Minimal Message Length [8] to determine it. We denote by $\hat{G}^s$ the estimated causal graph for the $s$-th individual. It was determined as follows: $\hat{G}_{ij}^s = 1$ if $|\hat{b}_{ij}^s| > 0.1$, and $\hat{G}_{ij}^s = 0$ if otherwise. Alternatively, one may use Wald test to examine significance of edges, as in [32].

In Figure 2(Upper), we reported the $F_1$ score to measure the accuracy of learned causal graphs. Specifically, sub-figure (a) shows the $F_1$ score (y-axis) for the number of groups $q = 2$, the sample size of each individual $ls = 20$, and the number of individuals $n = 60, 80, 100$ (x-axis), (b) for $q = 2, n = 100$, and $ls = 20, 40, 60$ (x-axis), (c) for $q = 3, ls = 20$, and $n = 60, 80, 100$ (x-axis), and (d) for $q = 3, n = 100$, and $ls = 20, 40, 60$ (x-axis). We can see that our proposed method SSCM has the best performance (the highest $F_1$) in all cases, and the accuracy slightly increases along with the number of individuals or the sample size per individual. MC, IB, and LiNGAM show similar performance and are less accurate than SSCM. MC and IB perform less well because they only take into account the first two orders of noise distributions. The performance of LiNGAM may be affected by the small sample size.

Besides the causal structure, which only takes into account the existence of an edge, we also compared the accuracy of the estimated causal strength quantitatively. It is important to compare the estimated causal strength, because in different groups the causal strength may be enhanced or inhibited while the causal structure remains the same. In particular, we compared the $L_2$ distance between the true causal strength and the estimation for each individual, i.e., $\|B^s - \hat{B}^s\|_2$. Figure 2(Lower)

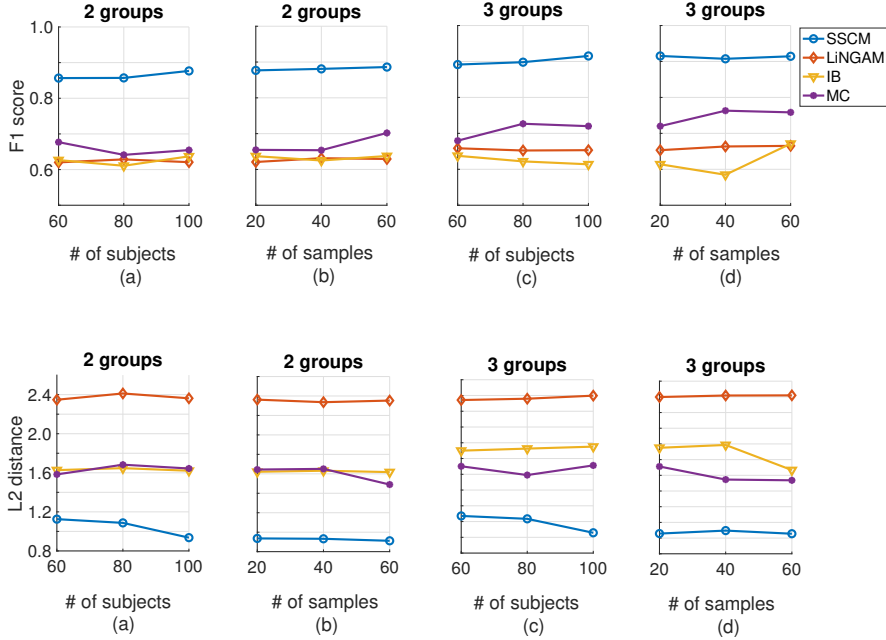

Figure 2: (Upper) F1 score of the recovered causal structure; (Lower) $L_2$ distance of the estimated causal strength.

reports the estimated $L_2$ distance with the proposed method in different settings, compared to that with LiNGAM, IB, and MC. Our SSCM gives the most accurate estimation of the causal strength (the smallest $L_2$ distance). MC and IB have the second-best accuracy, while LiNGAM performs less well. LiNGAM fails to estimate the quantitative causal strength accurately, although the estimated qualitative causal graph has a similar accuracy with MC and IB.

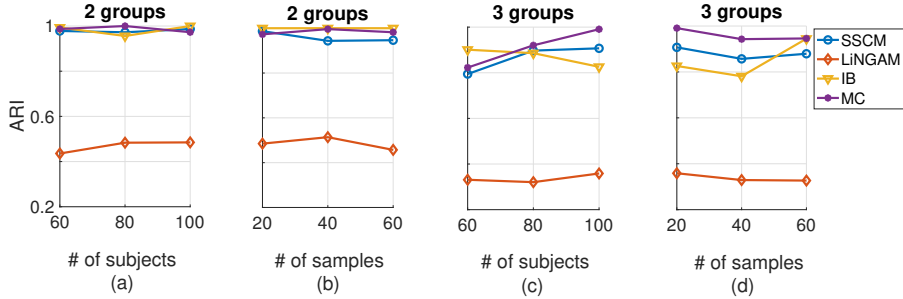

Figure 3: Adjusted Rand Index.

Next, we performed mechanism-based clustering by directly leveraging the estimated specific and shared causal model. Figure 3 gives the clustering performance in different settings, measured by Adjusted Rand Index (ARI [28]). It measures the similarity between the estimated groups and the ground truth (the higher, the more accurate). We compared our method with LiNGAM-K-Means, MC-K-Means, and IB-K-Means. LiNGAM-K-Means, MC-K-Means, and IB-K-Means use K-means to cluster the causal relations estimated by LiNGAM, MC, and IB, respectively. We also performed paired, one-sided Wilcoxon signed rank test on the estimated ARI between our method and each of the remaining ones [12], across different settings. Our method significantly outperforms LiNGAM, with $p$-values less than 0.002, and achieves performance that is comparable to MC and IB.

**fMRI Hippocampus** We applied our methods to the fMRI hippocampus data [27], which contains signals from six separate brain regions: perirhinal cortex (PRC), parahippocampal cortex (PHC),

entorhinal cortex (EC), subiculum (Sub), CA1, and CA3/Dentate Gyrus (CA3) in resting states on the same person on 84 successive days. We used anatomical connections [3, 39] as a reference. Some biological evidence has shown that in resting-states, the effective pathways in the hippocampus may change, depending on unmeasured intrinsic states [10]. We assume that the causal relations are fixed on the same day, but may change across different days. With the proposed method, we found that the causal relations between these six regions can be divided into two groups: in one group, the edge Sub → EC is inhibited; in the other group, EC → CA3 and CA1 → Sub are inhibited. This result is consistent with the finding that EC → CA3 and CA1 → Sub are usually involved in consolidation of a long-term memory [29], while Sub → EC is usually implicated in working memory [30]. The edge CA3 → CA1 is robust, existing in both groups, which coincides with the current findings in neuroscience [33].

Table 1: Clustering performance on flow cytometry data

| Methods | SSCM | LiNGAM | IB | MC | Plain K-Means |
|---------|------|--------|------|------|---------------|
| ARI | **0.92** | 0.21 | 0.78 | 0.25 | 0.87 |

**Cellular Signaling Networks**   We applied the proposed method to multivariate flow cytometry data, which were measured from 11 phosphorylated proteins and phospholipids [31]. A series of stimulatory cues and inhibitory interventions were performed, leading to different conditions. With different interventions in different conditions, the causal relations over the 11 variables may change across them. The data from each condition mimic a group, and in each condition, we segmented the data into subsets, with 30 samples in each subset, mimicking an individual. Table 1 reports the clustering performance, measured by ARI, on the data from condition `phorbol myristate acetate` and condition `anti-CD3 + anti-CD28 + LY294002`. Besides those comparisons in the simulation, we also compared the clustering performance with plain K-means, that is, directly applying K-means to the original data. Our method achieves the best performance, with ARI 0.92. Compared to the former condition, the causal strength of the following edges in the latter condition are inhibited: PIP2 → PIP3, Erk → Pka, Jnk → Pkc, and the following edges are enhanced: Raf → Mek, Mek → Raf, Akt → Pka, Pkc → P38. For the estimated cellular signaling networks under each condition, please see the supplementary material.

## 7   Conclusions and Future Work

In this paper, we proposed a unified framework for causal relations discovery and mechanism-based clustering. In particular, we developed a specific and shared causal model, which takes into account the variabilities of causal relations across individuals/groups and leverages commonalities to achieve statistically reliable estimation. Experimental results on synthetic and real-world data show that the learned SSCM gives the specific causal knowledge for each individual as well as the general trend over the population, and the estimated model directly provides the group information of each individual. Our current implementation relies on maximum likelihood estimation with SAEM, which does not generally scale well: currently we can handle 10 variables with 200 subjects within 1 hour. For the purpose of improving scalability, one line of our future work is to use likelihood-free frameworks for parameter estimation with, e.g., adversarial learning. Moreover, we will extend our methods to cover nonlinear causal relationships, to partially observable processes, and to data with selection bias [41].

## Acknowledgements

We thank Petar Stojanov for helping to revise the paper. We would like to acknowledge the support by National Institutes of Health under Contract No. NIH-1R01EB022858-01, FAIN-R01EB022858, NIH-1R01LM012087, NIH-5U54HG008540-02, and FAIN-U54HG008540, by the United States Air Force under Contract No. FA8650-17-C-7715, and by National Science Foundation EAGER Grant No. IIS-1829681. The National Institutes of Health, the U.S. Air Force, and the National Science Foundation are not responsible for the views reported in this article. KZ also benefited from funding from Living Analytics Research Center and Singapore Management University.

## Footnotes

*Correspondence to: Biwei Huang, email: `biweih@andrew.cmu.edu`

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
