[Supplementary Material]

# Supplementary Material for "Specific and Shared Causal Relation Modeling and Mechanism-Based Clustering"

**Biwei Huang** [1] *, **Kun Zhang**[1], **Pengtao Xie**[2], **Mingming Gong**[3], **Eric Xing**[2,4], **Clark Glymour**[1]

[1]Department of Philosophy, Carnegie Mellon University, Pittsburgh, PA, USA.
[2]Petuum Inc., USA.
[3]School of Mathematics and Statistics, University of Melbourne, Melbourne, Australia.
[4]Department of Machine Learning, Carnegie Mellon University, Pittsburgh, PA, USA.

## S1   Derivations in M step

In this section, we give detailed derivations of the M step in the SAEM algorithm. We consider two cases separately: with only instantaneous causal relations from i.i.d. data (Section S1.1), and with both instantaneous and time-lagged causal relations from stationary time series (Section S1.2).

### S1.1   With Instantaneous Causal Relations

With only instantaneous causal relations, the specific and shared causal model is represented as

$$X(t) = BX(t) + E(t), \tag{1}$$

with

$$
\begin{aligned}
P(b_{ij}) &= \sum_{k=1}^{q} \pi_k \mathcal{N}(b_{ij}|\mu_{k,ij}, \sigma_{k,ij}^2), \\
p(E) &= \sum_{k=1}^{q} \pi_k \sum_{k'=1}^{q'} \pi_{k,k'}^E \mathcal{N}(E|\mu_{k,k'}^E, \Sigma_{k,k'}^E),
\end{aligned}
\tag{2}
$$

where $B$ is an $m \times m$ causal adjacency matrix with entries $b_{ij}$.

The model defined in (1) and (2) can be regarded as a latent variable model, with $\mathbf{U} = \left\{ \{b_{ij}\}_{i,j=1}^m, \{a_{ij,p}\} \right\}$ as latent variables that we are interested in, and $\theta = \left\{ \{\pi_k\}, \{\mu_{k,ij}\}, \{\sigma_{k,ij}\}, \{\nu_{k,ij,p}\}, \{\omega_{k,ij,p}\}, \{\pi_{k,k'}^E\}, \{\mu_{k,k'}^E\}, \{\Sigma_{k,k'}^E\} \right\}$ as free parameters that need to be estimated. In particular, we exploit a stochastic approximation expectation maximization (SAEM [1]) algorithm, combined with Gibbs sampling in the E step, for model estimation.

SAEM computes the E step by Monte Carlo integration and uses a stochastic approximation update of the quantity $\mathcal{Q}$ at the $r$th iteration:

$$\tilde{\mathcal{Q}}_r(\theta) = (1 - \alpha_r)\tilde{\mathcal{Q}}_{r-1}(\theta) + \alpha_r \sum_{j=1}^{M} \frac{1}{M} \log P_\theta(\mathbf{X}^{1:n}, \mathring{U}^{(1:n,r,j)}), \tag{3}$$

where $\mathring{U}$ indicates sampled particles of $\mathbf{U}$, $M$ the generated number of particles, $\mathbf{X}^{1:n} = \{\mathbf{X}^s\}_{s=1}^n$ and $\mathbf{X}^s = (X^s(1), \cdots, X^s(l_s))$, $\mathring{U}^{(1:n,r,j)} = \{\mathring{U}^{(s,r,j)}\}_{s=1}^n$, and $\{\alpha_r\}_{r \geq 1}$ is a decreasing sequence of positive step size, with $\sum_r \alpha_r = \infty$ and $\sum_r \alpha_r^2 < \infty$.

By inductive reasoning, $\tilde{\mathcal{Q}}_r(\theta)$ can be rewritten as

$$\tilde{\mathcal{Q}}_r(\theta) = \frac{1}{M} \sum_{i=1}^{r} \sum_{j=1}^{M} (1 - \alpha_r)(1 - \alpha_{r-1}) \cdots (1 - \alpha_{i+1})\alpha_i \cdot L^{(i,j)}, \tag{4}$$

where $L^{(i,j)} = \log P_\theta(\mathbf{X}^{1:n}, B^{(1:n,i,j)})$. Let $\alpha^{(i)} = (1 - \alpha_r)(1 - \alpha_{r-1}) \cdots (1 - \alpha_{i+1})\alpha_i$. More specifically,

$$
\begin{aligned}
& \log P_\theta(\mathbf{X}^{1:n}, B^{(1:n,i,j)}) \\
=\ & \sum_{s=1}^{n} \log P_\theta(\mathbf{X}^s, B^{(s,i,j)}) \\
=\ & \sum_{s=1}^{n} \left( \log P_\theta(\mathbf{X}^{(s)}|B^{(s,i,j)}) + \log P_\theta(B^{(s,i,j)}) \right) \\
=\ & \sum_{s=1}^{n} \left( l_s \log |\det(I - B^{(s,i,j)})| + \sum_{t=1}^{l_s} \log P_\theta(E_t^{(s,i,j)}) + \log P_\theta(B^{(s,i,j)}) \right),
\end{aligned}
\tag{5}
$$

where

$$
P_\theta(E_t^{(s,i,j)}) = \sum_{k=1}^{q} \sum_{k'=1}^{q'} \pi_k \pi_{kk'}^{E} \mathcal{N}(E_t^{(s,i,j)}; \vec{\mu}_{kk'}^{E}, \Sigma_{kk'}^{E}),
\tag{6}
$$

and

$$
P_\theta(B^{(s,i,j)}) = \prod_{i_1 \neq i_2} P_\theta(b_{i_1 i_2}^{(s,i,j)}) = \prod_{i_1 \neq i_2} \sum_{k=1}^{q} \pi_k \mathcal{N}(b_{i_1 i_2}^{(s,i,j)} | \mu_{k,i_1 i_2}, \sigma_{k,i_1 i_2}^2).
\tag{7}
$$

For presentation convenience, we reorganize the form of some parameters and latent variables. Let $\tilde{B}$ be an $m(m-1) \times 1$ vector, which is derived by stacking each column of $B$ in sequence after removing diagonal entries. The same operation is applied to $\mu_{k,i_1 i_2}$ to get $\vec{\mu}$. $\Sigma_k$ is an $m(m-1) \times m(m-1)$ diagonal matrix, with entries $\sigma_{k,i_1 i_2}^2$. Thus,

$$
P_\theta(B^{(s,i,j)}) = P_\theta(\tilde{B}^{(s,i,j)}) = \sum_{k=1}^{q} \pi_k \mathcal{N}(\tilde{B}^{(s,i,j)} | \vec{\mu}_k, \Sigma_k).
\tag{8}
$$

Each parameter is estimated by setting the corresponding partial derivative of the expected log-likelihood $\tilde{\mathcal{Q}}_r$ to zero.

To estimate $\pi_{kk'}^{E}$, we add a regularization term on $\tilde{\mathcal{Q}}_r$ to guarantee $\sum_{k'=1}^{q'} \pi_{kk'}^{E} = 1$, and have

$$
\begin{aligned}
& \frac{\partial \tilde{\mathcal{Q}}_r + \lambda_k (\sum_{k'=1}^{q'} \pi_{kk'}^{E} - 1)}{\partial \pi_{kk'}^{E}} \\
=\ & \frac{1}{M} \sum_{i=1}^{r} \sum_{j=1}^{M} \alpha^{(i)} \sum_{s=1}^{n} \sum_{t=1}^{l_s} \frac{\partial \log P(E_t^{(s,i,j)})}{\partial \pi_{kk'}^{E}} + \frac{\partial \lambda_k (\sum_{k'=1}^{q'} \pi_{kk'} - 1)}{\partial \pi_{kk'}^{E}} \\
=\ & \frac{1}{M} \sum_{i=1}^{r} \sum_{j=1}^{M} \alpha^{(i)} \sum_{s=1}^{n} \sum_{t=1}^{l_s} \frac{\partial \log \sum_{k=1}^{q} \sum_{k'=1}^{q'} \pi_k \pi_{kk'}^{E} \mathcal{N}(E_t^{(s,i,j)}; \vec{\mu}_{kk'}^{E}, \Sigma_{kk'}^{E})}{\partial \pi_{kk'}^{E}} + \frac{\partial \lambda_k (\sum_{k'=1}^{q'} \pi_{kk'}^{E} - 1)}{\partial \pi_{kk'}^{E}} \\
=\ & \frac{1}{M} \sum_{i=1}^{r} \sum_{j=1}^{M} \alpha^{(i)} \sum_{s=1}^{n} \sum_{t=1}^{l_s} \frac{\pi_k \mathcal{N}(E_t^{(s,i,j)}; \vec{\mu}_{kk'}^{E}, \Sigma_{kk'}^{E})}{\sum_{k=1}^{q} \sum_{k'=1}^{q'} \pi_k \pi_{kk'}^{E} \mathcal{N}(E_t^{(s,i,j)}; \vec{\mu}_{kk'}^{E}, \Sigma_{kk'}^{E})} + \lambda_k.
\end{aligned}
\tag{9}
$$

Set $\frac{\partial \tilde{\mathcal{Q}}_r + \lambda_k (\sum_{k'=1}^{q'} \pi_{kk'}^{E} - 1)}{\partial \pi_{kk'}^{E}} = 0$, and multiply $\pi_{kk'}^{E}$ on both sides and sum over $k'$, and derive

$$
\frac{1}{M} \sum_{i=1}^{r} \sum_{j=1}^{M} \alpha^{(i)} \sum_{s=1}^{n} \sum_{t=1}^{l_s} \frac{\sum_{k'=1}^{q'} \pi_k \pi_{kk'}^{E} \mathcal{N}(E_t^{(s,i,j)}; \vec{\mu}_{kk'}^{E}, \Sigma_{kk'}^{E})}{\sum_{k=1}^{q} \sum_{k'=1}^{q'} \pi_k \pi_{kk'}^{E} \mathcal{N}(E_t^{(s,i,j)}; \vec{\mu}_{kk'}^{E}, \Sigma_{kk'}^{E})} + \lambda_k = 0,
\tag{10}
$$

and thus

$$
\lambda_k = -\frac{1}{M} \sum_{i=1}^{r} \sum_{j=1}^{M} \alpha^{(i)} \sum_{s=1}^{n} \sum_{t=1}^{l_s} \frac{\sum_{k'=1}^{q'} \pi_k \pi_{kk'}^{E} \mathcal{N}(E_t^{(s,i,j)}; \vec{\mu}_{kk'}^{E}, \Sigma_{kk'}^{E})}{\sum_{k=1}^{q} \sum_{k'=1}^{q'} \pi_k \pi_{kk'}^{E} \mathcal{N}(E_t^{(s,i,j)}; \vec{\mu}_{kk'}^{E}, \Sigma_{kk'}^{E})}.
\tag{11}
$$

Let $\gamma_{s,i,j,k,k',t}^{E} = \frac{\pi_k \pi_{kk'}^{E} \mathcal{N}(E_t^{(s,i,j)}; \vec{\mu}_{kk'}^{E}, \Sigma_{kk'}^{E})}{\sum_{k=1}^{q} \sum_{k'=1}^{q'} \pi_k \pi_{kk'}^{E} \mathcal{N}(E_t^{(s,i,j)}; \vec{\mu}_{kk'}^{E}, \Sigma_{kk'}^{E})}$, and thus

$$
\lambda_k = -\frac{1}{M} \sum_{i=1}^{r} \sum_{j=1}^{M} \alpha^{(i)} \sum_{s=1}^{n} \sum_{t=1}^{l_s} \sum_{k'=1}^{q'} \gamma_{s,i,j,k,k',t}^{E}.
\tag{12}
$$

We multiply $\pi_{kk'}^E$ on both sides, and derive

$$
\begin{aligned}
\hat{\pi}_{kk'}^E & \\
= & \frac{\frac{1}{M}\sum\limits_{i=1}^{r}\sum\limits_{j=1}^{M}\alpha^{(i)}\sum\limits_{s=1}^{n}\sum\limits_{t=1}^{l_s}\frac{\pi_k\pi_{kk'}^E\mathcal{N}(E_t^{(s,i,j)};\vec{\mu}_{kk'}^E,\Sigma_{kk'}^E)}{\Sigma_{k=1}^{q}\Sigma_{k'=1}^{q'}\pi_k\pi_{kk'}^E\mathcal{N}(E_t^{(s,i,j)};\vec{\mu}_{kk'}^E,\Sigma_{kk'}^E)}}{-\lambda_k} \\
= & \frac{\sum\limits_{i=1}^{r}\sum\limits_{j=1}^{M}\alpha^{(i)}\sum\limits_{s=1}^{n}\sum\limits_{t=1}^{l_s}\gamma_{s,i,j,k,k',t}^E}{\sum\limits_{i=1}^{r}\sum\limits_{j=1}^{M}\alpha^{(i)}\sum\limits_{s=1}^{n}\sum\limits_{t=1}^{l_s}\sum\limits_{k'=1}^{q'}\gamma_{s,i,j,k,k',t}^E}.
\end{aligned}
\tag{13}
$$

By taking the derivative of $\tilde{\mathcal{Q}}_k(\theta)$ w.r.t $\mu_{kk'}^E$, we have

$$
\begin{aligned}
\frac{\partial\tilde{\mathcal{Q}}_r}{\partial\mu_{kk'}^E} & \\
= & \frac{1}{M}\sum_{i=1}^{r}\sum_{j=1}^{M}\alpha^{(i)}\sum_{s=1}^{n}\sum_{t=1}^{l_s}\frac{\partial\log P(E_t^{(s,i,j)})}{\partial\mu_{kk'}^E} \\
= & \frac{1}{M}\sum_{i=1}^{r}\sum_{j=1}^{M}\alpha^{(i)}\sum_{s=1}^{n}\sum_{t=1}^{l_s}\frac{\partial\log\sum_{k=1}^{q}\sum_{k'=1}^{q'}\pi_k\pi_{kk'}^E\mathcal{N}(E_t^{(s,i,j)};\vec{\mu}_{kk'}^E,\Sigma_{kk'}^E)}{\partial\mu_{kk'}^E} \\
= & \frac{1}{M}\sum_{i=1}^{r}\sum_{j=1}^{M}\alpha^{(i)}\sum_{s=1}^{n}\sum_{t=1}^{l_s}\frac{\pi_k\pi_{kk'}^E\mathcal{N}(E_t^{(s,i,j)};\vec{\mu}_{kk'}^E,\Sigma_{kk'}^E)}{\sum_{k=1}^{q}\sum_{k'=1}^{q'}\pi_k\pi_{kk'}^E\mathcal{N}(E_t^{(s,i,j)};\vec{\mu}_{kk'}^E,\Sigma_{kk'}^E)}\cdot{\Sigma_{kk'}^E}^{-1}(E_t^{(s,i,j)}-\mu_{kk'}^E) \\
= & \frac{1}{M}\sum_{i=1}^{r}\sum_{j=1}^{M}\alpha^{(i)}\sum_{s=1}^{n}\sum_{t=1}^{l_s}\gamma_{s,i,j,k,k',t}^E\cdot{\Sigma_{kk'}^E}^{-1}(E_t^{(s,i,j)}-\mu_{kk'}^E).
\end{aligned}
\tag{14}
$$

Set $\frac{\partial\tilde{\mathcal{Q}}_r}{\partial\mu_{kk'}^E}=0$, and thus

$$
\hat{\mu}_{kk'}^E = \frac{\sum\limits_{i=1}^{r}\sum\limits_{j=1}^{M}\alpha^{(i)}\sum\limits_{s=1}^{n}\sum\limits_{t=1}^{l_s}\gamma_{s,i,j,k,k',t}^E\cdot E_t^{(s,i,j)}}{\sum\limits_{i=1}^{r}\sum\limits_{j=1}^{M}\alpha^{(i)}\sum\limits_{s=1}^{n}\sum\limits_{t=1}^{l_s}\gamma_{s,i,j,k,k',t}^E}.
\tag{15}
$$

By taking the derivative of $\tilde{\mathcal{Q}}_k(\theta)$ w.r.t $\Sigma_{kk'}^E$, we have

$$
\begin{aligned}
\frac{\partial\tilde{\mathcal{Q}}_r}{\partial\Sigma_{kk'}^E} & \\
= & \frac{1}{M}\sum_{i=1}^{r}\sum_{j=1}^{M}\alpha^{(i)}\sum_{s=1}^{n}\sum_{t=1}^{l_s}\frac{\partial\log P(E_t^{(s,i,j)})}{\partial\Sigma_{kk'}^E} \\
= & \frac{1}{M}\sum_{i=1}^{r}\sum_{j=1}^{M}\alpha^{(i)}\sum_{s=1}^{n}\sum_{t=1}^{l_s}\frac{\partial\log\sum_{k=1}^{q}\sum_{k'=1}^{q'}\pi_k\pi_{kk'}^E\mathcal{N}(E_t^{(s,i,j)};\vec{\mu}_{kk'}^E,\Sigma_{kk'}^E)}{\partial\Sigma_{kk'}^E} \\
= & \frac{1}{M}\sum_{i=1}^{r}\sum_{j=1}^{M}\alpha^{(i)}\sum_{s=1}^{n}\sum_{t=1}^{l_s}\frac{\pi_k\pi_{kk'}^E\mathcal{N}(E_t^{(s,i,j)};\vec{\mu}_{kk'}^E,\Sigma_{kk'}^E)}{\sum_{k=1}^{q}\sum_{k'=1}^{q'}\pi_k\pi_{kk'}^E\mathcal{N}(E_t^{(s,i,j)};\vec{\mu}_{kk'}^E,\Sigma_{kk'}^E)} \\
& \cdot\left(\frac{1}{2}{\Sigma_{kk'}^E}^{-1}(E_t^{(s,i,j)}-\mu_{kk'}^E)(E_t^{(s,i,j)}-\mu_{kk'}^E)^{\mathrm{T}}{\Sigma_{kk'}^E}^{-1}-\frac{1}{2}{\Sigma_{kk'}^E}^{-1}\right) \\
= & \frac{1}{M}\sum_{i=1}^{r}\sum_{j=1}^{M}\alpha^{(i)}\sum_{s=1}^{n}\sum_{t=1}^{l_s}\gamma_{s,i,j,k,k',t}^E\cdot\left(\frac{1}{2}{\Sigma_{kk'}^E}^{-1}(E_t^{(s,i,j)}-\mu_{kk'}^E)(E_t^{(s,i,j)}-\mu_{kk'}^E)^{\mathrm{T}}{\Sigma_{kk'}^E}^{-1}-\frac{1}{2}{\Sigma_{kk'}^E}^{-1}\right).
\end{aligned}
\tag{16}
$$

Set $\frac{\partial\tilde{\mathcal{Q}}_r}{\partial\Sigma_{kk'}^E}=0$, and thus

$$
\hat{\Sigma}_{kk'}^E = \frac{\sum\limits_{i=1}^{r}\sum\limits_{j=1}^{M}\alpha^{(i)}\sum\limits_{s=1}^{n}\sum\limits_{t=1}^{l_s}\gamma_{s,i,j,k,k',t}^E\cdot(E_t^{(s,i,j)}-\mu_{kk'}^E)(E_t^{(s,i,j)}-\mu_{kk'}^E)^{\mathrm{T}}}{\sum\limits_{i=1}^{r}\sum\limits_{j=1}^{M}\alpha^{(i)}\sum\limits_{s=1}^{n}\sum\limits_{t=1}^{l_s}\gamma_{s,i,j,k,k',t}^E}.
\tag{17}
$$

To estimate $\pi_k$, we add a regularization term on $\tilde{\mathcal{Q}}_r$ to guarantee $\sum_{k=1}^{q} \pi_k = 1$, and get

$$
\frac{\partial \tilde{\mathcal{Q}}_r + \lambda(\sum_k \pi_k - 1)}{\partial \pi_k}
$$

$$
= \frac{1}{M} \sum_{i=1}^{r} \sum_{j=1}^{M} \alpha^{(i)} \sum_{s=1}^{n} \frac{\partial \sum_{t=1}^{l_s} \log P_\theta(E_t^{(s,i,j)}) + \log P_\theta(B^{(s,i,j)})}{\partial \pi_k} + \lambda
$$

$$
= \frac{1}{M} \sum_{i=1}^{r} \sum_{j=1}^{M} \alpha^{(i)} \sum_{s=1}^{n} \left( \sum_{t=1}^{l_s} \frac{\sum_{k'=1}^{q'} \pi_{kk'}^E \mathcal{N}(E_t^{(s,i,j)}; \vec{\mu}_{kk'}^E, \Sigma_{kk'}^E)}{\sum_{k=1}^{q} \sum_{k'=1}^{q'} \pi_k \pi_{kk'}^E \mathcal{N}(E_t^{(s,i,j)}; \vec{\mu}_{kk'}^E, \Sigma_{kk'}^E)} + \frac{\mathcal{N}(\tilde{B}^{(s,i,j)}; \vec{\mu}_k, \Sigma_k)}{\sum_{k=1}^{q} \pi_k \mathcal{N}(\tilde{B}^{(s,i,j)}; \vec{\mu}_k, \Sigma_k)} \right) + \lambda.
$$
(18)

Set $\frac{\partial \tilde{\mathcal{Q}}_r + \lambda(\sum_k \pi_k - 1)}{\partial \pi_k} = 0$, multiply both sides by $\pi_k$ and sum over $k$:

$$
\frac{1}{M} \sum_{i=1}^{r} \sum_{j=1}^{M} \alpha^{(i)} \sum_{s=1}^{n} \cdot (l_s + 1) + \lambda = 0,
$$
(19)

and thus

$$
\lambda = -\frac{1}{M} \sum_{i=1}^{r} \sum_{j=1}^{M} \alpha^{(i)} \sum_{s=1}^{n} \cdot (l_s + 1).
$$
(20)

Multiply both sides by $\pi_k$:

$$
\frac{1}{M} \sum_{i=1}^{r} \sum_{j=1}^{M} \alpha^{(i)} \sum_{s=1}^{n} \left( \sum_{t=1}^{l_s} \frac{\sum_{k'=1}^{q'} \pi_k \pi_{kk'}^E \mathcal{N}(E_t^{(s,i,j)}; \vec{\mu}_{kk'}^E, \Sigma_{kk'}^E)}{\sum_{k=1}^{q} \sum_{k'=1}^{q'} \pi_k \pi_{kk'}^E \mathcal{N}(E_t^{(s,i,j)}; \vec{\mu}_{kk'}^E, \Sigma_{kk'}^E)} + \frac{\pi_k \mathcal{N}(\tilde{B}^{(s,i,j)}; \vec{\mu}_{kk'}^E, \Sigma_{kk'}^E)}{\sum_{k=1}^{q} \pi_k \mathcal{N}(\tilde{B}^{(s,i,j)}; \vec{\mu}_k, \Sigma_k)} \right) + \lambda \pi_k = 0.
$$
(21)

Let

$$
\gamma_{s,i,j,k} = \frac{\pi_k \mathcal{N}(\tilde{B}^{(s,i,j)}; \vec{\mu}_{k'}, \Sigma_{k'})}{\sum_{k=1}^{q} \pi_k \mathcal{N}(\tilde{B}^{(s,i,j)}; \vec{\mu}_k, \Sigma_k)},
$$

so

$$
\hat{\pi}_k = \frac{\sum_{i=1}^{r} \sum_{j=1}^{M} \alpha^{(i)} \sum_{s=1}^{n} \left( \sum_{t=1}^{l_s} \sum_{k'=1}^{q'} \gamma_{s,i,j,k,k',t}^E + \gamma_{s,i,j,k} \right)}{\sum_{i=1}^{r} \sum_{j=1}^{M} \alpha^{(i)} \sum_{s=1}^{n} (l_s + 1)}.
$$
(22)

By taking the derivative of $\tilde{\mathcal{Q}}_k(\theta)$ w.r.t $\vec{\mu}_k$, we have

$$
\frac{\partial \tilde{\mathcal{Q}}_r}{\partial \vec{\mu}_k}
$$

$$
= \frac{1}{M} \sum_{i=1}^{r} \sum_{j=1}^{M} \alpha^{(i)} \sum_{s=1}^{n} \frac{\partial \log P(B^{(s,i,j)})}{\partial \vec{\mu}_k}
$$

$$
= \frac{1}{M} \sum_{i=1}^{r} \sum_{j=1}^{M} \alpha^{(i)} \sum_{s=1}^{n} \frac{\partial \log \sum_{k=1}^{q} \pi_k \mathcal{N}(\tilde{B}^{(s,i,j)}; \vec{\mu}_k, \Sigma_k)}{\partial \vec{\mu}_k}
$$
(23)

$$
= \frac{1}{M} \sum_{i=1}^{r} \sum_{j=1}^{M} \alpha^{(i)} \sum_{s=1}^{n} \frac{\pi_k \mathcal{N}(\tilde{B}^{(s,i,j)}; \vec{\mu}_k, \Sigma_k)}{\sum_{k=1}^{q} \pi_k \mathcal{N}(\tilde{B}^{(s,i,j)}; \vec{\mu}_k, \Sigma_k)} \cdot \Sigma_k^{-1} (\tilde{B}^{(s,i,j)} - \vec{\mu}_k)
$$

$$
= \frac{1}{M} \sum_{i=1}^{r} \sum_{j=1}^{M} \alpha^{(i)} \sum_{s=1}^{n} \gamma_{s,i,j,k} \cdot \Sigma_k^{-1} (\tilde{B}^{(s,i,j)} - \vec{\mu}_k).
$$

Set $\frac{\partial \tilde{\mathcal{Q}}_r}{\partial \vec{\mu}_k} = 0$, and thus

$$
\hat{\vec{\mu}}_k = \frac{\sum_{i=1}^{r} \sum_{j=1}^{M} \alpha^{(i)} \sum_{s=1}^{n} \gamma_{s,i,j,k} \cdot \tilde{B}^{(s,i,j)}}{\sum_{i=1}^{r} \sum_{j=1}^{M} \alpha^{(i)} \sum_{s=1}^{n} \gamma_{s,i,j,k}}.
$$
(24)

By taking the derivative of $\tilde{\mathcal{Q}}_k(\theta)$ w.r.t $\Sigma_k$, we have

$$
\begin{aligned}
\frac{\partial \tilde{\mathcal{Q}}_r}{\partial \Sigma_k} \\
= \quad & \frac{1}{M} \sum_{i=1}^{r} \sum_{j=1}^{M} \alpha^{(i)} \sum_{s=1}^{n} \frac{\partial \log P(B^{(s,i,j)})}{\partial \Sigma_k} \\
= \quad & \frac{1}{M} \sum_{i=1}^{r} \sum_{j=1}^{M} \alpha^{(i)} \sum_{s=1}^{n} \frac{\partial \log \sum_{k=1}^{q} \pi_k \mathcal{N}(\tilde{B}^{(s,i,j)}; \vec{\mu}_k, \Sigma_k)}{\partial \Sigma_k} \\
= \quad & \frac{1}{M} \sum_{i=1}^{r} \sum_{j=1}^{M} \alpha^{(i)} \sum_{s=1}^{n} \frac{\pi_k \mathcal{N}(\tilde{B}^{(s,i,j)}; \vec{\mu}_k, \Sigma_k)}{\sum_{k=1}^{q} \pi_k \mathcal{N}(\tilde{B}^{(s,i,j)}; \vec{\mu}_k, \Sigma_k)} \cdot \left( \tfrac{1}{2} \Sigma_k^{-1} (\tilde{B}^{(s,i,j)} - \vec{\mu}_k)(\tilde{B}^{(s,i,j)} - \vec{\mu}_k)^{\mathrm{T}} \Sigma_k^{-1} - \tfrac{1}{2} \Sigma_k^{-1} \right) \\
= \quad & \frac{1}{M} \sum_{i=1}^{r} \sum_{j=1}^{M} \alpha^{(i)} \sum_{s=1}^{n} \gamma_{s,i,j,k} \cdot \left( \tfrac{1}{2} \Sigma_k^{-1} (\tilde{B}^{(s,i,j)} - \vec{\mu}_k)(\tilde{B}^{(s,i,j)} - \vec{\mu}_k)^{\mathrm{T}} \Sigma_k^{-1} - \tfrac{1}{2} \Sigma_k^{-1} \right).
\end{aligned}
\tag{25}
$$

Set $\frac{\partial \tilde{\mathcal{Q}}_r}{\partial \Sigma_k} = 0$, and thus

$$
\hat{\Sigma}_k = \frac{\sum_{i=1}^{r} \sum_{j=1}^{M} \alpha^{(i)} \sum_{s=1}^{n} \gamma_{s,i,j,k} \cdot (\tilde{B}^{(s,i,j)} - \vec{\mu}_k)(\tilde{B}^{(s,i,j)} - \vec{\mu}_k)^{\mathrm{T}}}{\sum_{i=1}^{r} \sum_{j=1}^{M} \alpha^{(i)} \sum_{s=1}^{n} \gamma_{s,i,j,k}}.
\tag{26}
$$

Therefore, we update the parameters in the inner EM of the M step with the following way:

$$
\begin{aligned}
\hat{\vec{\mu}}_k &= \frac{\sum_{i=1}^{r} \sum_{j=1}^{M} \alpha^{(i)} \sum_{s=1}^{n} \gamma_{s,i,j,k} \cdot \tilde{B}^{(s,i,j)}}{\sum_{i=1}^{r} \sum_{j=1}^{M} \alpha^{(i)} \sum_{s=1}^{n} \gamma_{s,i,j,k}}, \\
\hat{\Sigma}_k &= \frac{\sum_{i=1}^{r} \sum_{j=1}^{M} \alpha^{(i)} \sum_{s=1}^{n} \gamma_{s,i,j,k} \cdot (\tilde{B}^{(s,i,j)} - \vec{\mu}_k)(\tilde{B}^{(s,i,j)} - \vec{\mu}_k)^{\mathrm{T}}}{\sum_{i=1}^{r} \sum_{j=1}^{M} \alpha^{(i)} \sum_{s=1}^{n} \gamma_{s,i,j,k}}, \\
\hat{\pi}_k &= \frac{\sum_{i=1}^{r} \sum_{j=1}^{M} \alpha^{(i)} \sum_{s=1}^{n} \left( \sum_{t=1}^{l_s} \sum_{k'=1}^{q'} \gamma_{s,i,j,k,k',t}^{E} + \gamma_{s,i,j,k} \right)}{\sum_{i=1}^{r} \sum_{j=1}^{M} \alpha^{(i)} \sum_{s=1}^{n} (l_s + 1)}, \\
\hat{\mu}_{kk'}^{E} &= \frac{\sum_{i=1}^{r} \sum_{j=1}^{M} \alpha^{(i)} \sum_{s=1}^{n} \sum_{t=1}^{l_s} \gamma_{s,i,j,k,k',t}^{E} \cdot E_t^{(s,i,j)}}{\sum_{i=1}^{r} \sum_{j=1}^{M} \alpha^{(i)} \sum_{s=1}^{n} \sum_{t=1}^{l_s} \gamma_{s,i,j,k,k',t}^{E}}, \\
\hat{\Sigma}_{kk'}^{E} &= \frac{\sum_{i=1}^{r} \sum_{j=1}^{M} \alpha^{(i)} \sum_{s=1}^{n} \sum_{t=1}^{l_s} \gamma_{s,i,j,k,k',t}^{E} \cdot (E_t^{(s,i,j)} - \mu_{kk'}^{E})(E_t^{(s,i,j)} - \mu_{kk'}^{E})^{\mathrm{T}}}{\sum_{i=1}^{r} \sum_{j=1}^{M} \alpha^{(i)} \sum_{s=1}^{n} \sum_{t=1}^{l_s} \gamma_{s,i,j,k,k',t}^{E}}, \\
\hat{\pi}_{kk'} &= \frac{\sum_{i=1}^{r} \sum_{j=1}^{M} \alpha^{(i)} \sum_{s=1}^{n} \sum_{t=1}^{l_s} \gamma_{s,i,j,k,k',t}^{E}}{\sum_{i=1}^{r} \sum_{j=1}^{M} \alpha^{(i)} \sum_{s=1}^{n} \sum_{t=1}^{l_s} \sum_{k'=1}^{q'} \gamma_{s,i,j,k,k',t}^{E}},
\end{aligned}
\tag{27}
$$

where

$$
\gamma_{s,i,j,k} = \frac{\pi_k \mathcal{N}(\tilde{B}^{(s,i,j)}; \vec{\mu}_k, \Sigma_k)}{\sum_{k=1}^{q} \pi_k \mathcal{N}(\tilde{B}^{(s,i,j)}; \vec{\mu}_k, \Sigma_k)},
$$

and

$$
\gamma_{s,i,j,k,k',t}^{E} = \frac{\pi_k \pi_{kk'}^{E} \mathcal{N}(E_t^{(s,i,j)}; \vec{\mu}_{kk'}^{E}, \Sigma_{kk'}^{E})}{\sum_{k=1}^{q} \sum_{k'=1}^{q'} \pi_k \pi_{kk'}^{E} \mathcal{N}(E_t^{(s,i,j)}; \vec{\mu}_{kk'}^{E}, \Sigma_{kk'}^{E})}.
$$

## S1.2  With Both Instantaneous and Time-Lagged Causal Relations

With both instantaneous and time-lagged causal relations, the specific and shared causal model is represented as

$$
X(t) = BX(t) + \sum_{p=1}^{p_l} A_p X(t-p) + E(t),
\tag{28}
$$

with

$$
\begin{aligned}
P(b_{ij}) &= \sum_{k=1}^{q} \pi_k \mathcal{N}(b_{ij}|\mu_{k,ij}, \sigma_{k,ij}^2), \\
P(a_{ij,p}) &= \sum_{k=1}^{q} \pi_k \mathcal{N}(a_{ij,p}|\nu_{k,ij,p}, \omega_{k,ij,p}^2), \\
p(E) &= \sum_{k=1}^{q} \pi_k \sum_{k'=1}^{q'} \pi_{k,k'}^E \mathcal{N}(E|\mu_{k,k'}^E, \Sigma_{k,k'}^E).
\end{aligned}
\tag{29}
$$

The quantity $\mathcal{Q}$ at the $r$th iteration in SAEM is:

$$
\tilde{\mathcal{Q}}_r(\theta) = (1 - \alpha_r)\hat{\mathcal{Q}}_{r-1}(\theta) + \frac{1}{M}\alpha_r \sum_{j=1}^{M} \log P_\theta(\mathbf{X}^{1:n}, B^{(1:n,r,j)}, A^{(1:n,r,j)}),
\tag{30}
$$

with

$$
\begin{aligned}
&\log P_\theta(\mathbf{X}^{1:n}, B^{(1:n,i,j)}, A^{(1:n,i,j)}) \\
=\ & \sum_{s=1}^{n} \log P_\theta(\mathbf{X}^s, B^{(s,i,j)}, A^{(s,i,j)}) \\
=\ & \sum_{s=1}^{n} \left( \log P_\theta(\mathbf{X}^{(s)}|B^{(s,i,j)}, A^{(s,i,j)}) + \log P_\theta(B^{(s,i,j)}) + \log P_\theta(A^{(s,i,j)}) \right) \\
=\ & \sum_{s=1}^{n} \left( l_s \log |\det(I - B^{(s,i,j)})| + \sum_{t=1}^{l_s} \log P_\theta(E_t^{(s,i,j)}) + \log P_\theta(B^{(s,i,j)}) + \log P_\theta(A^{(s,i,j)}) \right),
\end{aligned}
\tag{31}
$$

where

$$
P_\theta(E_t^{(s,i,j)}) = \sum_{k=1}^{q} \sum_{k'=1}^{q'} \pi_k \pi_{kk'}^E \mathcal{N}(E_t^{(s,i,j)}; \vec{\mu}_{kk'}^E, \Sigma_{kk'}^E),
\tag{32}
$$

and

$$
P_\theta(B^{(s,i,j)}) = \prod_{i_1 \neq i_2} P_\theta(b_{i_1 i_2}^{(s,i,j)}) = \prod_{i_1 \neq i_2} \sum_{k=1}^{q} \pi_k \mathcal{N}(b_{i_1 i_2}^{(s,i,j)}|\mu_{k,i_1 i_2}, \sigma_{k,i_1 i_2}^2),
\tag{33}
$$

and

$$
P_\theta(A^{(s,i,j)}) = \prod_{i_1,i_2} P_\theta(a_{i_1 i_2}^{(s,i,j)}) = \prod_{i_1,i_2} \sum_{k=1}^{q} \pi_k \mathcal{N}(a_{i_1 i_2}^{(s,i,j)}|\nu_{k,i_1 i_2}, \omega_{k,i_1 i_2}^2).
\tag{34}
$$

For presentation convenience, we reorganize the form of some parameters and latent variables. Let $\tilde{B}$ be an $m(m-1) \times 1$ vector, which is derived by stacking each column of $B$ in sequence after removing diagonal entries. The same operation is applied to $\mu_{k,i_1 i_2}$ to get $\vec{\mu}$, to $\nu$ to derive $\vec{\nu}$, and to $A$ to derive $\tilde{A}$. $\Sigma_k$ is an $m(m-1) \times m(m-1)$ diagonal matrix, with entries $\sigma_{k,i_1 i_2}^2$. Similarly, we can derive $\Omega_k$. Thus,

$$
P_\theta(B^{(s,i,j)}) = P_\theta(\tilde{B}^{(s,i,j)}) = \sum_{k=1}^{q} \pi_k \mathcal{N}(\tilde{B}^{(s,i,j)}|\vec{\mu}_k, \Sigma_k),
\tag{35}
$$

and

$$
P_\theta(A^{(s,i,j)}) = P_\theta(\tilde{A}^{(s,i,j)}) = \sum_{k=1}^{q} \pi_k \mathcal{N}(\tilde{A}^{(s,i,j)}|\vec{\nu}_k, \Omega_k).
\tag{36}
$$

By inductive reasoning, $\tilde{\mathcal{Q}}_r(\theta)$ can be rewritten as

$$
\tilde{\mathcal{Q}}_r(\theta) = \frac{1}{M} \sum_{i=1}^{r} \sum_{j=1}^{M} (1 - \alpha_r)(1 - \alpha_{r-1}) \cdots (1 - \alpha_{i+1})\alpha_i \cdot L^{(i,j)},
\tag{37}
$$

where $L^{(i,j)} = \log P_\theta(\mathbf{X}^{1:n}, B^{(1:n,i,j)}, A^{(1:n,i,j)})$. Let $\alpha^{(i)} = (1-\alpha_r)(1-\alpha_{r-1}) \cdots (1-\alpha_{i+1})\alpha_i$.

Each parameter is estimated by setting the corresponding partial derivative of the expected log-likelihood $\tilde{\mathcal{Q}}_r$ to zero.

The estimations of $\pi_{kk'}$, $\mu_{kk'}^E$, $\Sigma_{kk'}^E$, $\mu_k$, and $\Sigma_k$ are the same as those in Section S1.1.

To estimate $\pi_k$, we add a regularization term on $\tilde{\mathcal{Q}}_r$ to guarantee $\sum_{k=1}^q \pi_k = 1$, and have

$$
\begin{aligned}
&\frac{\partial \tilde{\mathcal{Q}}_r + \lambda(\sum_k \pi_k - 1)}{\partial \pi_k} \\
=\ & \frac{1}{M} \sum_{i=1}^r \sum_{j=1}^M \alpha^{(i)} \sum_{s=1}^n \frac{\partial \sum_{t=1}^{l_s} \log P_\theta(E_t^{(s,i,j)}) + \log P_\theta(B^{(s,i,j)}) + \log P_\theta(A^{(s,i,j)})}{\partial \pi_k} + \lambda \\
=\ & \frac{1}{M} \sum_{i=1}^r \sum_{j=1}^M \alpha^{(i)} \sum_{s=1}^n \left( \sum_{t=1}^{l_s} \frac{\sum_{k'=1}^{q'} \pi_{kk'}^E \mathcal{N}(E_t^{(s,i,j)}; \vec{\mu}_{kk'}^E, \Sigma_{kk'}^E)}{\sum_{k=1}^q \sum_{k'=1}^{q'} \pi_k \pi_{kk'}^E \mathcal{N}(E_t^{(s,i,j)}; \vec{\mu}_{kk'}^E, \Sigma_{kk'}^E)} + \frac{\mathcal{N}(\tilde{B}^{(s,i,j)}; \vec{\mu}_k, \Sigma_k)}{\sum_{k=1}^q \pi_k \mathcal{N}(\tilde{B}^{(s,i,j)}; \vec{\mu}_k, \Sigma_k)} \right. \\
& \left. + \frac{\mathcal{N}(\tilde{A}^{(s,i,j)}; \vec{\nu}_k, \Omega_k)}{\sum_{k=1}^q \pi_k \mathcal{N}(\tilde{A}^{(s,i,j)}; \vec{\nu}_k, \Omega_k)} \right) + \lambda.
\end{aligned}
\tag{38}
$$

Set $\frac{\partial \tilde{\mathcal{Q}}_r + \lambda(\sum_k \pi_k - 1)}{\partial \pi_k} = 0$, multiply both sides by $\pi_k$ and sum over $k$:

$$
\frac{1}{M} \sum_{i=1}^r \sum_{j=1}^M \alpha^{(i)} \sum_{s=1}^n \cdot (l_s + 1 + 1) + \lambda = 0,
\tag{39}
$$

and thus

$$
\lambda = -\frac{1}{M} \sum_{i=1}^r \sum_{j=1}^M \alpha^{(i)} \sum_{s=1}^n \cdot (l_s + 2).
\tag{40}
$$

Multiply both sides by $\pi_k$:

$$
\begin{aligned}
&\frac{1}{M} \sum_{i=1}^r \sum_{j=1}^M \alpha^{(i)} \sum_{s=1}^n \left( \sum_{t=1}^{l_s} \frac{\sum_{k'=1}^{q'} \pi_k \pi_{kk'}^E \mathcal{N}(E_t^{(s,i,j)}; \vec{\mu}_{kk'}^E, \Sigma_{kk'}^E)}{\sum_{k=1}^q \sum_{k'=1}^{q'} \pi_k \pi_{kk'}^E \mathcal{N}(E_t^{(s,i,j)}; \vec{\mu}_{kk'}^E, \Sigma_{kk'}^E)} + \frac{\pi_k \mathcal{N}(\tilde{B}^{(s,i,j)}; \vec{\mu}_{kk'}^E, \Sigma_{kk'}^E)}{\sum_{k=1}^q \pi_k \mathcal{N}(\tilde{B}^{(s,i,j)}; \vec{\mu}_k, \Sigma_k)} \right. \\
& \left. + \frac{\pi_k \mathcal{N}(\tilde{A}^{(s,i,j)}; \vec{\nu}_{kk'}^E, \Omega_{kk'}^E)}{\sum_{k=1}^q \pi_k \mathcal{N}(\tilde{A}^{(s,i,j)}; \vec{\nu}_k, \Omega_k)} \right) + \lambda \pi_k = 0.
\end{aligned}
\tag{41}
$$

Let

$$
\gamma_{s,i,j,k}^B = \frac{\pi_k \mathcal{N}(\tilde{B}^{(s,i,j)}; \vec{\mu}_{k'}, \Sigma_{k'})}{\sum_{k=1}^q \pi_k \mathcal{N}(\tilde{B}^{(s,i,j)}; \vec{\mu}_k, \Sigma_k)},
$$

and

$$
\gamma_{s,i,j,k}^A = \frac{\pi_k \mathcal{N}(\tilde{A}^{(s,i,j)}; \vec{\nu}_{k'}, \Omega_{k'})}{\sum_{k=1}^q \pi_k \mathcal{N}(\tilde{A}^{(s,i,j)}; \vec{\nu}_k, \Omega_k)},
$$

so

$$
\hat{\pi}_k = \frac{\sum_{i=1}^r \sum_{j=1}^M \alpha^{(i)} \sum_{s=1}^n \left( \sum_{t=1}^{l_s} \sum_{k'=1}^{q'} \gamma_{s,i,j,k,k',t}^E + \gamma_{s,i,j,k}^B + \gamma_{s,i,j,k}^A \right)}{\sum_{i=1}^r \sum_{j=1}^M \alpha^{(i)} \sum_{s=1}^n (l_s + 2)}.
\tag{42}
$$

By taking the derivative of $\tilde{\mathcal{Q}}_k(\theta)$ w.r.t $\vec{\nu}_k$, we have

$$
\begin{aligned}
&\frac{\partial \tilde{\mathcal{Q}}_r}{\partial \vec{\nu}_k} \\
=\ & \frac{1}{M} \sum_{i=1}^r \sum_{j=1}^M \alpha^{(i)} \sum_{s=1}^n \frac{\partial \log P(A^{(s,i,j)})}{\partial \vec{\nu}_k} \\
=\ & \frac{1}{M} \sum_{i=1}^r \sum_{j=1}^M \alpha^{(i)} \sum_{s=1}^n \frac{\partial \log \sum_{k=1}^q \pi_k \mathcal{N}(\tilde{A}^{(s,i,j)}; \vec{\nu}_k, \Omega_k)}{\partial \vec{\nu}_k} \\
=\ & \frac{1}{M} \sum_{i=1}^r \sum_{j=1}^M \alpha^{(i)} \sum_{s=1}^n \frac{\pi_k \mathcal{N}(\tilde{A}^{(s,i,j)}; \vec{\nu}_k, \Omega_k)}{\sum_{k=1}^q \pi_k \mathcal{N}(\tilde{A}^{(s,i,j)}; \vec{\nu}_k, \Omega_k)} \cdot \Omega_k^{-1} (\tilde{A}^{(s,i,j)} - \vec{\nu}_k) \\
=\ & \frac{1}{M} \sum_{i=1}^r \sum_{j=1}^M \alpha^{(i)} \sum_{s=1}^n \gamma_{s,i,j,k} \cdot \Omega_k^{-1} (\tilde{A}^{(s,i,j)} - \vec{\nu}_k).
\end{aligned}
\tag{43}
$$

Set $\frac{\partial \tilde{\mathcal{Q}}_r}{\partial \vec{\nu}_k} = 0$, and thus

$$
\hat{\vec{\nu}}_k = \frac{\sum_{i=1}^r \sum_{j=1}^M \alpha^{(i)} \sum_{s=1}^n \gamma_{s,i,j,k}^A \cdot \tilde{A}^{(s,i,j)}}{\sum_{i=1}^r \sum_{j=1}^M \alpha^{(i)} \sum_{s=1}^n \gamma_{s,i,j,k}^A}.
\tag{44}
$$

By taking the derivative of $\tilde{\mathcal{Q}}_k(\theta)$ w.r.t $\Omega_k$, we have

$$
\begin{aligned}
&\frac{\partial \tilde{\mathcal{Q}}_r}{\partial \Omega_k} \\
=\ & \frac{1}{M}\sum_{i=1}^{r}\sum_{j=1}^{M}\alpha^{(i)}\sum_{s=1}^{n}\frac{\partial \log P(A^{(s,i,j)})}{\partial \Omega_k} \\
=\ & \frac{1}{M}\sum_{i=1}^{r}\sum_{j=1}^{M}\alpha^{(i)}\sum_{s=1}^{n}\frac{\partial \log \sum_{k=1}^{q}\pi_k\mathcal{N}(\tilde{A}^{(s,i,j)};\vec{\nu}_k,\Omega_k)}{\partial \Omega_k} \\
=\ & \frac{1}{M}\sum_{i=1}^{r}\sum_{j=1}^{M}\alpha^{(i)}\sum_{s=1}^{n}\frac{\pi_k\mathcal{N}(\tilde{A}^{(s,i,j)};\vec{\nu}_k,\Omega_k)}{\sum_{k=1}^{q}\pi_k\mathcal{N}(\tilde{A}^{(s,i,j)};\vec{\nu}_k,\Omega_k)}\cdot\left(\tfrac{1}{2}\Omega_k^{-1}(\tilde{A}^{(s,i,j)}-\vec{\nu}_k)(\tilde{A}^{(s,i,j)}-\vec{\nu}_k)^{\mathrm{T}}\Omega_k^{-1}-\tfrac{1}{2}\Omega_k^{-1}\right) \\
=\ & \frac{1}{M}\sum_{i=1}^{r}\sum_{j=1}^{M}\alpha^{(i)}\sum_{s=1}^{n}\gamma_{s,i,j,k}^{A}\cdot\left(\tfrac{1}{2}\Omega_k^{-1}(\vec{A}^{(s,i,j)}-\vec{\nu}_k)(\tilde{A}^{(s,i,j)}-\vec{\nu}_k)^{\mathrm{T}}\Omega_k^{-1}-\tfrac{1}{2}\Omega_k^{-1}\right).
\end{aligned}
\tag{45}
$$

Set $\frac{\partial \tilde{\mathcal{Q}}_r}{\partial \Omega_k}=0$, and thus

$$
\hat{\Omega}_k=\frac{\sum_{i=1}^{r}\sum_{j=1}^{M}\alpha^{(i)}\sum_{s=1}^{n}\gamma_{s,i,j,k}^{A}\cdot(\tilde{A}^{(s,i,j)}-\vec{\nu}_k)(\tilde{A}^{(s,i,j)}-\vec{\nu}_k)^{\mathrm{T}}}{\sum_{i=1}^{r}\sum_{j=1}^{M}\alpha^{(i)}\sum_{s=1}^{n}\gamma_{s,i,j,k}^{A}}.
\tag{46}
$$

Therefore, we update the parameters in the inner EM of the M step with the following way:

$$
\begin{aligned}
\hat{\vec{\mu}}_k &= \frac{\sum_{i=1}^{r}\sum_{j=1}^{M}\alpha^{(i)}\sum_{s=1}^{n}\gamma_{s,i,j,k}^{B}\cdot\tilde{B}^{(s,i,j)}}{\sum_{i=1}^{r}\sum_{j=1}^{M}\alpha^{(i)}\sum_{s=1}^{n}\gamma_{s,i,j,k}^{B}}, \\
\hat{\Sigma}_k &= \frac{\sum_{i=1}^{r}\sum_{j=1}^{M}\alpha^{(i)}\sum_{s=1}^{n}\gamma_{s,i,j,k}\cdot(\tilde{B}^{(s,i,j)}-\vec{\mu}_k)(\tilde{B}^{(s,i,j)}-\vec{\mu}_k)^{\mathrm{T}}}{\sum_{i=1}^{r}\sum_{j=1}^{M}\alpha^{(i)}\sum_{s=1}^{n}\gamma_{s,i,j,k}^{B}}, \\
\hat{\vec{\nu}}_k &= \frac{\sum_{i=1}^{r}\sum_{j=1}^{M}\alpha^{(i)}\sum_{s=1}^{n}\gamma_{s,i,j,k}^{A}\cdot\tilde{A}^{(s,i,j)}}{\sum_{i=1}^{r}\sum_{j=1}^{M}\alpha^{(i)}\sum_{s=1}^{n}\gamma_{s,i,j,k}^{A}}, \\
\hat{\Omega}_k &= \frac{\sum_{i=1}^{r}\sum_{j=1}^{M}\alpha^{(i)}\sum_{s=1}^{n}\gamma_{s,i,j,k}^{A}\cdot(\tilde{A}^{(s,i,j)}-\vec{\nu}_k)(\tilde{A}^{(s,i,j)}-\vec{\nu}_k)^{\mathrm{T}}}{\sum_{i=1}^{r}\sum_{j=1}^{M}\alpha^{(i)}\sum_{s=1}^{n}\gamma_{s,i,j,k}^{A}}, \\
\hat{\pi}_k &= \frac{\sum_{i=1}^{r}\sum_{j=1}^{M}\alpha^{(i)}\sum_{s=1}^{n}\left(\sum_{t=1}^{l_s}\sum_{k'=1}^{q'}\gamma_{s,i,j,k,k',t}^{E}+\gamma_{s,i,j,k}^{B}+\gamma_{s,i,j,k}^{A}\right)}{\sum_{i=1}^{r}\sum_{j=1}^{M}\alpha^{(i)}\sum_{s=1}^{n}(l_s+2)}, \\
\hat{\mu}_{kk'}^{E} &= \frac{\sum_{i=1}^{r}\sum_{j=1}^{M}\alpha^{(i)}\sum_{s=1}^{n}\sum_{t=1}^{l_s}\gamma_{s,i,j,k,k',t}^{E}\cdot E_t^{(s,i,j)}}{\sum_{i=1}^{r}\sum_{j=1}^{M}\alpha^{(i)}\sum_{s=1}^{n}\sum_{t=1}^{l_s}\gamma_{s,i,j,k,k',t}^{E}}, \\
\hat{\Sigma}_{kk'}^{E} &= \frac{\sum_{i=1}^{r}\sum_{j=1}^{M}\alpha^{(i)}\sum_{s=1}^{n}\sum_{t=1}^{l_s}\gamma_{s,i,j,k,k',t}^{E}\cdot(E_t^{(s,i,j)}-\mu_{kk'}^{E})(E_t^{(s,i,j)}-\mu_{kk'}^{E})^{\mathrm{T}}}{\sum_{i=1}^{r}\sum_{j=1}^{M}\alpha^{(i)}\sum_{s=1}^{n}\sum_{t=1}^{l_s}\gamma_{s,i,j,k,k',t}^{E}}, \\
\hat{\pi}_{kk'} &= \frac{\sum_{i=1}^{r}\sum_{j=1}^{M}\alpha^{(i)}\sum_{s=1}^{n}\sum_{t=1}^{l_s}\gamma_{s,i,j,k,k',t}^{E}}{\sum_{i=1}^{r}\sum_{j=1}^{M}\alpha^{(i)}\sum_{s=1}^{n}\sum_{t=1}^{l_s}\sum_{k'=1}^{q'}\gamma_{s,i,j,k,k',t}^{E}},
\end{aligned}
\tag{47}
$$

where

$$
\gamma_{s,i,j,k}^{B}=\frac{\pi_k\mathcal{N}(\tilde{B}^{(s,i,j)};\vec{\mu}_k,\Sigma_k)}{\sum_{k=1}^{q}\pi_k\mathcal{N}(\tilde{B}^{(s,i,j)};\vec{\mu}_k,\Sigma_k)},
$$

$$
\gamma_{s,i,j,k}^{A}=\frac{\pi_k\mathcal{N}(\tilde{A}^{(s,i,j)};\vec{\nu}_k,\Omega_k)}{\sum_{k=1}^{q}\pi_k\mathcal{N}(\tilde{A}^{(s,i,j)};\vec{\nu}_k,\Omega_k)},
$$

and

$$\gamma^E_{s,i,j,k,k',t} = \frac{\pi_k \pi^E_{kk'} \mathcal{N}(E^{(s,i,j)}_t; \vec{\mu}^E_{kk'}, \Sigma^E_{kk'})}{\sum_{k=1}^{q} \sum_{k'=1}^{q'} \pi_k \pi^E_{kk'} \mathcal{N}(E^{(s,i,j)}_t; \vec{\mu}^E_{kk'}, \Sigma^E_{kk'})}.$$

## S2    Proof of Theorem 1

In order to prove Theorem 1, we first give the following lemma about identifiability of mixture models from grouped samples.

**Lemma S1** (Identifiability of Mixture Models from Grouped Samples [6]). *Suppose we have observations from a mixture model and that they are grouped, such that observations in the same group are known to be drawn from the same component. Denote by $q$ the number of groups. If there are at least $2q - 1$ observations per group, any mixture of $q$ probability measures can be uniquely identified.*

*Proof.* See [6]. □

Next, we give the proof of Theorem 1, based on Lemma S1.

*Proof.* Condition 1 means that $b_{ij}$ and $a_{ij,p}$ take a degenerate Gaussian distribution in each group; their distributions can be represented as follows:

$$P(b_{ij}) = \sum_{k=1}^{q} \pi_k \delta_{\mu_{k,ij}}(b_{ij}), \quad P(a_{ij,p}) = \sum_{k=1}^{q} \pi_k \delta_{\nu_{k,ij,p}}(a_{ij,p}),$$

where $\delta_{\mu_{k,ij}}(b_{ij}) = 1$, if $b_{ij} = \mu_{k,ij}$, and 0 if otherwise; $\delta_{\nu_{k,ij,p}}(a_{ij,p}) = 1$, if $a_{ij} = \nu_{k,ij,p}$, and 0 if otherwise. With condition 1, the identifiability of the proposed causal model can be seen as finite mixture models with grouped samples. The "grouped samples" means that for each individual there are several samples, and it is known in advance that they are identically distributed samples from the same component. Note the difference between identifiability of finite mixture models with grouped samples and the case where the observations are drawn i.i.d. from a mixture model.

Under condition 1, where the parameters $\sigma_{k,ij} = 0$ and $\omega_{k,ij,p} = 0$, for all $i, j, k, p \in \mathcal{N}^+$, the proposed causal model can be seen as finite mixture models with grouped samples. Furthermore, according to Lemma S1, if condition 2 ($l_s > 2q - 1$) satisfies, the cumulative distribution function of each mixture component, as well as the mixture proportion, is identifiable; that is, $P(X|z_k = 1)$ is identifiable, for $k = 1, \cdots, q$, and $P(Z)$ is identifiable.

After identifying $P(X|z_k = 1)$, for $k = 1, \cdots, q$, we next show that instantaneous causal relations $b_{ij}$ and lagged causal relations $a_{ij,p}$ in each group are identifiable. Because $b_{ij}$ and $a_{ij,p}$ are fixed within the group, individuals at group $k$ satisfy the following generating process:

$$X(t) = BX(t) + \sum_{p=1}^{p_l} A_p X(t - p) + E(t), \tag{48}$$

where $B$ and $A_p$ are free parameters. The above equation can be reorganized as

$$X(t) = \sum_{p=1}^{p_l} (I - B)^{-1} A_p X(t - p) + (I - B)^{-1} E(t). \tag{49}$$

It has been show both $B$ and $A_p$ in Eq. (49) are identifiable and their estimations are consistent, given that the noise term $E$ is non-Gaussian and that the instantaneous causal structure for each individual is acyclic [2].

In addition, if there are cycles in the instantaneous causal structure, the identifiability requires two more conditions [3]: (1) the cycles are disjoint, and (2) the causal model is stable, i.e., $\lim_{k \to \infty} B^k = 0$.

□

## S3  Cellular Signaling Networks

We applied the proposed method to multivariate flow cytometry data, which were measured from 11 phosphorylated proteins and phospholipids [5]. The 11 variables are `Raf`, `Mek`, `Plc`, `PIP2`, `PIP3`, `Erk`, `Akt`, `Pka`, `Pkc`, `P38`, and `Jnk`. A series of stimulatory cues and inhibitory interventions were performed, leading to different conditions. With different interventions in different conditions, the causal relations over the 11 variables may change across them. The data from each condition mimic a group, and in each condition, we segmented the data into subsets, with 30 samples in each subset, mimicking an individual.

We applied SSCM on the data from condition `phorbol myristate acetate` (denoted by condition 1) and condition `anti-CD3 + anti-CD28 + LY294002` (denoted by condition 2). After learning the model parameters, we clustered the subsets (individuals) into groups. Table S1 reports the clustering performance, measured by Adjusted Rand Index (ARI [4]). Our method achieves the best performance, with ARI 0.92.

Table S1: Clustering performance on flow cytometry data

| Methods | SSCM | LiNGAM | IB | MC | Plain K-Means |
|---|---|---|---|---|---|
| ARI | **0.92** | 0.21 | 0.78 | 0.25 | 0.87 |

We then estimated group-specific causal graphs. We denoted by $\hat{B}^s$ the estimated causal adjacency matrix for group (condition) $s$, with $s = 1, 2$. Figure S1 shows the difference between the estimated causal adjacency matrix from group 1 and that from group 2, i.e., $\hat{B}^1 - \hat{B}^2$. We can see that compared to group 1, the causal strength of the following edges in group 2 are inhibited: PIP2 $\rightarrow$ PIP3, Erk $\rightarrow$ Pka, Jnk $\rightarrow$ Pkc, and the following edges are enhanced: Raf $\rightarrow$ Mek, Mek $\rightarrow$ Raf, Akt $\rightarrow$ Pka, Pkc $\rightarrow$ P38. The findings are consistent with the performed interventions. In condition 2, reagents anti-CD3, anti-CD28, and LY294002 are used. Specifically, anti-CD3/CD28 activates T cells and induces proliferation and cytokine production. Induced signaling through the T cell receptor activates Plc, Raf, Mek, Erk, and Pkc, while LY294002 activates Akt and inhibits PIP2 [5]. From our results, we found that Pkc, Raf, Mek, and Akt were activated, and PIP2 were inhibited.

Figure S1: The difference between the estimated causal adjacency matrix from condition 1 and that from condition 2, i.e., $\hat{B}^1 - \hat{B}^2$.

Furthermore, we denoted by $\hat{G}^s$ the estimated causal graph of group $s$, with $s = 1, 2$. It was determined as follows: $\hat{G}^s_{ij} = 1$ if $|\hat{b}^s_{ij}| > 0.2$, and $\hat{G}^s_{ij} = 0$ if otherwise. Figure S2(a) and (b) give the estimated cellular signaling networks of group 1 and group 2, respectively.

## Footnotes

*Correspondence to: Biwei Huang, email: `biweih@andrew.cmu.edu`.