[Reviews · NeurIPS 2019]

Reviewer 1



In many scenarios, the causal relationships considered over a set of variables vary across groups and at the same time share some common causal relationships. So it is better to find different causal graphs for each individual. This paper solves this problem by first dividing the set of agents into a number of groups and then finding a causal graph for each group. The authors propose a model over m variables that includes both instantaneous effects and time-lagged effects. Ideally, we would have to estimate this model separately for each user, but that might be impossible with a small number of samples. So the model assumes a mixture of Gaussian prior for the effects. The number of components in the mixture is the number of clusters and the goal is to estimate the prior probabilities over the clusters and individual components of the mixture. Subsequently, the authors use EM algorithm to estimate the parameters of the model. However, computing the posterior exactly is intractable, so they use Monte Carlo integration and stochastic approximation for the E step. Both the simulation and experiment on real-world dataset show that the proposed method performs better than various existing methods in terms of F1 score, clustering, and approximating the true model. I have some suggestions and questions for the authors: 1. Theorem 1 proves an identification result for the degenerate distributions. What breaks down for the general case even if we consider just the instantaneous effects? It would have been nice to see a discussion in this regard. 2. How did the authors choose the parameter l_p the number of time-steps considered for time-lagged effects? 3. Why did the authors choose the threshold of 0.1 for converting the weights to the presence / absence of edges in the graph? Is there a systematic procedure to guide this choice? Originality This paper makes a significant contribution in terms of proposing a new model for sharing causal relations. The proposed algorithm seems to recover individual specific causal graphs and will be of immense interest to researchers working in the field if it can be scaled across a large number of clusters and a large number of variables. Quality I thought the paper makes several significant contributions and will be really helpful for the researchers working in the field of causal modeling and causal discovery. Clarity I thoroughly enjoyed reading the paper. Both the model and the experiment section was clear to me. However, I thought that the paper might benefit from a brief discussion of SAEM algorithm before deriving the steps of the algorithm. Significance The modeling contributions of this paper are sound. The proposed algorithm is interesting, seems to perform better than the existing methods and will be significant if it can be scaled for a large number of variables and a large number of clusters.

Reviewer 2



Update after author rebuttal: I greatly appreciate the authors responding point-by-point to my concerns. I have updated my score to a 7. If the paper is ultimately accepted, in addition to the clarifications the authors made in the rebuttal, I stress that the paper would be improved by clarifying the following 1) The point about direction switching. I am not from a neuroscience/biology background (I believe most of the NeurIPS community is not either) and so the justification here is a little counterintuitive here. The authors might want to consider contrasting this work with existing ways one might (inadequately) represent such behavior (e.g. latent confounders when Gaussianity/linearity is not assumed) -- this could be an elaboration of the comments in the rebuttal. 2) From the rebuttal: "In our simulations under non-zero variance settings, we never observed that the procedure converged to wrong solutions, suggesting that the non-zero-variance case is also identifiable" -- just because a sim didn't obtain non-identifiability, doesn't mean the model is identified. I think the authors' intuition is likely to be correct, however they should consider softening the language around this point. -------------------------------------------------------------------------- This paper is well-written. There are some points the authors should clarify in review and in a future draft to enhance the manuscript. Details of my personal confusions below: 0) What is the relation between the present work and the causal relational modeling literature (e.g. Arbour, Marazopoulou, and Jensen 16, Marazopoulou, Maier, and Jensen 15, and other works from David Jensen and associates)? Is "relational" used in the same sense here? 1) In general, how is this approach different from learning a DAG-based model where there is a cluster indicator variable that indexes the rest of the graph? In my mind this would help with statistical concerns (i.e. you can "lump" all subjects together) to learn one graph rather than several graphs (one per group). 2) In the abstract: "...due to possibly omitted factors that affect the quantitative causal effects." To be clear this refers _only_ to missing edges and not unobserved variables (e.g. latent confounders) instead/as well? 3) "in healthcare, individuals may show different responses to the same treatment". In line with #1 above, this seems more like a distributional notion rather than needing a model that learns separate graphs for separate groups, no? Existing causal effect estimation (more or less developed than discovery is up for debate) tends to be more suited for handling a single non-parametric graph for the purpose of identification and then placing assumptions after identification has been obtained. Would it be valuable for discovery methods to try to follow this approach (at least in the vein of learning a single unified graph that represents the distribution of subjects)? 4) "In addition, they do not allow opposite causal directions..." Following #1 and #3 this seems like a weird thing to allow for in the proposed model. Can the authors please clarify why they permit edge direction to switch among subjects? I'm not familiar with areas in the causal literature that have considered this case and as is, the justification seems light. 5) Sec 2 paragraph 1 (and Conclusion) -- I agree that adding a treatment of partial observations ("hidden confounders" etc) seems challenging. Can the authors please give a high level discussion of the steps they might take towards allowing that behavior in their model? 6) Eq. 1: what's p_l in the middle summation? It doesn't seem to be defined 7) "However, the limited sample size from each individual limits statistical efficiency or even makes discovery impossible" -- is there a formal characterization/proof of this? I understand that statistical identification is challenging. E.g as n -> 0 you have a tougher and tougher time but is there a formal characterization of "impossibility" in this setting? 8) Below Thm 1: "we allow that across different groups, some causal directions are reversed" -- along with earlier comments, how does this provide more information about the ground truth graph than, say, learning an equivalence class that leave direction unspecified when it's unclear? 9) "(1) the cycles are disjoint" -- what does "disjoint" mean here? Why is this requirement necessary? 10) "our empirical results strongly suggest that the causal model is also identifiable" -- To me, this isn't sufficient for the same reason that association isn't sufficient for effect estimation contexts. Can the authors give justification that their experiments (non-sims) use data that satisfies the theoretical requirements implied by Thm 1? 11) "Imagine an extreme case: if there are enough samples ..." Why is is the case that this is identifiable? Can the authors please formalize this? 12) l_s > 2q - 1 -- What's the intuition behind this bound? How is it used in the proof of theorem 1? E.g. "We first show..." -- it seems the authors cite Vandermeulen and Scott 2015 rather than proving this is a sufficient bound in their setting. 13) The proof of theorem 1 seems incomplete. There doesn't appear to be a complete, formal proof in the appendix. It's not obvious from the cited papers that the same conclusions (from those citations) will hold in this setting. Can the authors please give a formal proof? 14) In Sec. 4.1, what is being adapted from SAEM and what is a new contribution? It seems the general EM is from SAEM and the Gibbs sampling procedure is novel? 15) "The computational complexity (...) is O(m^2 n M T0). This is very slow, especially since it is on a per-iteration level. Can the authors give a characterization of types of data where it would be feasible to run this procedure? Obviously the experiments in the penultimate section provide an example but it's not clear what scale of data they used there (e.g. was a very small subset of the total available fMRI data used and would a neuroscientist need to be able to use this procedure on a data set many orders of magnitude larger?) Additionally, how can one be sure that the Gibbs procedure has converged? 16) "It is east to add prior knowledge of causal connections" -- how is this achieved? An imposed independence or dependence constraint in Eq. 1? 17) For Gibbs and other procedures that require parameter initializations, is there some characterization available of how reliant these procedures' performance is on those initializations? 18) "We randomly generated acyclic causal structures according to the Erdos-Renyi model" -- ER gives undirected graphs. How did the authors choose directions of edges and ensure acyclicity in this sim? 19) "Other parameters were set as follows. sigma^2_{k, i, j} = U(.01, .1)" In theorem 1 the authors state that the distributions should be no variance -- why does this sim use distributions with non-zero variance? The authors should consider adding a sim that exactly matches the conditions that are understood according to theory. 20) fMRI and Cellular Signaling Networks data: is this non-gaussian data? How do we know that it is (non-Gaussian)? 21) fMRI "We assume that the causal relations are fixed on the same day, but may change across different days" -- Can the authors provide some insight from neuroscience that backs up this assumption? To me, it seems a little odd that the brain would be re-wired and electric flows would switch from one day to the next. Is there a well understood mechanism that explains this behavior? 22) Cellular: "With different interventions in different conditions, the causal relations over the 11 variables may change across them" -- Unless these are different cells in the different conditions, it would seem more reasonable to model this as having unobserved confounding rather than having causal direction switch according to interventions. Can the authors provide justification for this construction?

Reviewer 3



[Originality] The use of Gaussian mixture as a specific model for causal discovery from data with group-wise causal mechanisms seems novel and interesting. [Quality] - Considering that the essence of the paper is the proposal of using a mixture of Gaussians, experimental assessment on real-world data is quite important. - I am not entirely convinced by the experiments of the current version of the paper. (Table 1) Without a comparison against plain clustering methods (e.g., k-means), it seems still possible for SSCM to take advantage of other clustering signals such as distinct data regions, not the structural difference. To validate the proposed model, I think it is crucial for the paper to collect convincing evidence that the proposed method really likely conducted a mechanism-based clustering. For example, (1) showing the results of a comparison against a plain clustering method or (2) showing variability of the estimated graphs for each group (to see if the posterior is concentrated well around the MAP or the posterior mean) or (3) providing an interpretation of the estimated graphs based on domain knowledge (similarly to the one in the fMRI experiment) may help. Using a biased sampling from each group to create mock "individuals" may also be an option. - The problem of estimating the number of groups remains to be addressed. It seems to be an essentially difficult problem, but the paper did not specify a concrete method for it, and only used the underlying truth value for the experiments (line 262). [Clarity] - I think the manuscript is very well prepared. All paragraphs are easy and smooth to comprehend. - The only problem I had with the presentation is in the statement of Theorem 1. The notion of identifiability is often under some form of (hidden) asymptotics. If I understood correctly, for the case of Theorem 1, the identifiability is under the limit of $n \to \infty$. I think it is important to clarify, especially when there is a "sample size" in the statement of the theorem which is quite confusing (because in standard estimation problems, the identifiability of a parameter is under the limit of (sample size) \to \infty). - (Typo) Supplementary material p.9: "Adjusted Random Index" should be "Adjusted Rand Index." [Significance] Considering the nature of the paper (proposing a specific model), its significance largely depends on the experimental results using real-world data. The experimental results are interesting, providing some insights into the proposed model, but not completely satisfactory for the reasons stated above in the Quality section.

[Author Response · NeurIPS 2019]

We thank all the reviewers for the valuable insights and feedback. Below please see our response to the questions.

**Rev #1:** **(1)** *Brief description of SAEM:* Thank you for the suggestion. We will add a description before presenting the algorithm: "SAEM uses a stochastic approximation procedure to estimate the conditional expectation of the complete log-likelihood. More specifically, given the learned parameters in the current iteration, the values of latent variables are first sampled under the posteriori density. Then these sampled data are used to update the value of the conditional expectation of the complete log-likelihood with stochastic approximation." **(2)** $p_l$ *and threshold:* We use cross-validation to choose the optimal time lag: we compare the averaged $Q$ values over 10 iterations after convergence to choose $p_l$ ($Q$ values have small variations across iterations due to stochastic approximation). Regarding the threshold, 0.1 and 0.05 gave almost identical performance, and we reported the results with 0.1. We will include the results with Wald test to examine significance of edges, as in [24]. **(3)** Despite much time devoted to it, it does not seem feasible to have a concise proof for general cases in the near future, because it is hard to characterize identifiability in a Bayesian model.

**Rev #2: (Questions 23, 4, 8)** Causal direction flipping is not an assumption. We note that our model can handle more general cases, even the causal direction flips (line 143). For instance, in the brain network, different directions may be activated across subjects or states. It is hard to handle with traditional methods. **(24, 14)** We borrowed the framework of SAEM. The Gibbs sampling procedure in the E step and the derivations in the M step are our new contributions. We will make it clear. **(25, 12, 13)** Condition $l_s > 2q - 1$ is used to establish the identifiability of $P(X|z_k = 1)$ and $p(Z)$. As our model is a specific case of that by Vandermeulen & Scott (2015), we adapted their results. Following your suggestion, we will provide a complete proof in the supplements. **(26)** The current implementation is feasible for small- or median-scale systems (e.g., the 11-variable cellular network). As future work, we hope likelihood-free frameworks for parameter estimation with, e.g., adversarial learning, can improve the scalability. **(27, 18, 19)** In our implementation, skeletons and directions are generated in one step. Specifically, the graph $G$ (including directions) is generated as follows: $G = triu(ones(m), 1). * binornd(1, p, m, m), G = PGP^T$, where $binornd$ is a random number generator for binomial distributions, $P$ is a permutation matrix, and $G_{i,j} = 1$ means there is an edge from node $i$ to node $j$. The proposed SSCM does cover the case of non-zero variance, but currently the identifiability proof is only shown in a specific case. In our simulations under non-zero variance settings, we never observed that the procedure converged to wrong solutions, suggesting that the non-zero-variance case is also identifiable. Following your suggestion, we will also include simulations with zero variances. **(28, 20, 21, 22, 10)** Non-Gaussianity can be checked by "normality test." For the fMRI and cellular data, the null hypothesis was rejected at significance level 0.01. Regarding causal structure variation, for fMRI data, it is well-known that neural connectivities may change across different external stimuli or intrinsic states. Hippocampus is activated in resting state, working on different tasks, such as consolidation of episodic, autobiographical, or declarative memory, depending on unmeasured intrinsic states. In different recording days, hippocampus may focus on different tasks, leading to nonconstant causal mechanisms [2,11,22,25]. For cellular data, causal structure may be different across conditions/interventions. **(0)** They are different. Our work focuses on propositional data and uses functional causal models to represent causal relationships. A well-known graphical model that uses propositional representation is the Bayesian network. Jensen et al.'s work focuses on relational data, using a relational schema to specify types of entities, relationships, and attributes. **(1, 3)** The difference is that our method can capture structure differences across groups/subjects and can provide both personalized and shared graphs, which are essential for many tasks. **(2)** "Omitted factors" refers to unobserved variables which affect the causal effects between observed variables. **(5)** In light of previous identifiability result, we are able to extend the current method to allow confounders, by making use of the identifiability of over-complete ICA. **(6)** $p_l$ is the maximum time lag. **(9)** "Disjoint" means that the cycles do not interact with each other (i.e., no variable is involved in more than one cycle). Under this assumption, we can uniquely identify the causal graph. **(15)** The fMRI data we used contain 6 main regions in hippocampus. The Gibbs procedure is considered as convergent if the correlation between successive samples is smaller than a threshold. **(16)** If we know some edges are not possible, we can fix corresponding entries of $A$ or $B$ to 0. **(17)** If one randomly initializes all values, the F1 score is around 0.06 less, compared to the initialization given in lines 259-262, so the performance depends on initialization but not heavily. This will be made explicit.

**Rev #3: (1)** Yes. It is straightforward to extend it to more general forms. We will revise it, following your suggestion. **(2)** Thanks for the valuable comments. Yes. It should be $n \to \infty$. We will revise it in line 138: "... is identifiable, as $n \to \infty$, under the following conditions". **(3)** To be clearer, we can alternatively view SSCM as two separate steps: i) learning the causal structure of each subject and ii) clustering with the learned structure. Moreover, from Eqs. 13 and 14, we can see that the estimation of $P(X^s|z_k)$ depends only on causal adjacency matrices $A$ and $B$. Thus, SSCM performs clustering based on only structure. Following your suggestion, we directly applied K-means on the data and the clustering accuracy is 0.87, lower than that by SSCM. In contrast, K-means performs clustering based on the data distribution, not the causal influences. The dataset we used is obtained under different interventions, and we know that intervention may break some causal influences, so the structure is expected to differ across conditions, making it sensible to use structure information for clustering. We will include the result of K-means and the variability for estimated graphs in each group. **Others:** Regarding the number of groups, a naive way is to use cross validation, and using the Silhouette score with close clusters merged may be a better solution. This will be briefly discussed.

[Meta-Review · NeurIPS 2019]

The authors develop a "specific and shared causal model (SSCM)", and consider causal inference where a group-specific causal model is learned. While reviewers had a few specific issues that they raised, they all found the paper well-written, felt it was a good contribution to the existing causal discovery literature, and that it ought to be published in NeurIPS.